# Size-adaptive Hypothesis Testing for Fairness

**Antonio Ferrara** ⬤
CENTAI, Turin, Italy
Graz University of Technology, Austria
`antonio.ferrara@centai.eu`

**Francesco Cozzi** ⬤
Sapienza University, Rome, Italy
CENTAI, Turin, Italy
`f.cozzi@uniroma1.it`

**Alan Perotti** ⬤
CENTAI, Turin, Italy
`alan.perotti@centai.eu`

**André Panisson** ⬤
CENTAI, Turin, Italy
`panisson@centai.eu`

**Francesco Bonchi** ⬤
CENTAI, Turin, Italy
EURECAT, Barcelona, Spain
`bonchi@centai.eu`

## Abstract

Determining whether an algorithmic decision-making system discriminates against a specific demographic typically involves comparing a single point estimate of a fairness metric against a predefined threshold. This practice is statistically brittle: it ignores sampling error and treats small demographic subgroups the same as large ones. The problem intensifies in intersectional analyses, where multiple sensitive attributes are considered jointly, giving rise to a larger number of smaller groups. As these groups become more granular, the data representing them becomes too sparse for reliable estimation, and fairness metrics yield excessively wide confidence intervals, precluding meaningful conclusions about potential unfair treatments.

In this paper, we introduce a unified, size-adaptive, hypothesis-testing framework that turns fairness assessment into an evidence-based statistical decision. Our contribution is twofold. (i) For sufficiently large subgroups, we prove a Central-Limit result for the statistical parity difference, leading to analytic confidence intervals and a Wald test whose type-I (false positive) error is guaranteed at level $\alpha$. (ii) For the long tail of small intersectional groups, we derive a fully Bayesian Dirichlet–multinomial estimator; Monte-Carlo credible intervals are calibrated for any sample size and naturally converge to Wald intervals as more data becomes available. We validate our approach empirically on benchmark datasets, demonstrating how our tests provide interpretable, statistically rigorous decisions under varying degrees of data availability and intersectionality.

## 1 Introduction

As machine learning (ML) systems play an increasingly central role in decision-making in consequential domains – ranging from education and hiring to healthcare and criminal justice – concerns over algorithmic fairness have gained prominence, prompting the development of various fairness metrics [19, 29, 12, 41] and intervention strategies [46, 24]. Early proposals framed the problem as ensuring *statistical parity* across protected demographic groups defined by a single sensitive attribute such as race or gender. Demographic Parity [19], Equalized Odds and Equal Opportunity [29], Predictive Parity [12], and related criteria remain cornerstones of the field and are implemented in popular toolkits (e.g., FAIRLEARN [44]). These metrics are ordinarily reported pointwise, and the decision to classify an observed metric's value as a potential issue is addressed by defining a threshold: everything below this threshold is ignored, while everything above this threshold is treated as an issue [4]. Such a threshold-based approach ignores the statistical uncertainty inherent in estimating population parameters from finite samples, treating small and large populations equivalently [36].

Minor disparities in large groups might unjustly be overlooked, while larger disparities in smaller groups may not provide sufficient statistical evidence for meaningful conclusions [23, 33].

The issue of uncertainty of fairness measures is further exacerbated by *intersectionality* – an essential concept indicating the unique disadvantages faced by individuals belonging to multiple protected groups [14]. In intersectional settings, numerous subgroups emerge as multiple sensitive attributes are considered jointly. As these groups become more granular, the data representing them becomes too sparse for reliable estimation, and fairness metrics yield excessively wide confidence intervals, precluding meaningful conclusions about potential unfairness [45, 23]. This phenomenon produces a sort of *"resolution limit" of intersectionality*: as we show in our empirical assessment (Section 4) using the standard datasets commonly employed in fairness research, even the intersection of just two sensitive attributes can create conditions where existing methodologies appear to detect unfair treatment, when the uncertainty on the adopted metric is actually so large to make any unfairness claim statistically not significant.

This observation prompts a critical research question: *At what granularity should fairness be rigorously guaranteed, and how can we decide this in a principled, data-driven manner?* Specifically, there is a need for principled methodologies that identify a meaningful balance between the comprehensiveness of intersectionality and the statistical robustness of fairness assessments [30, 34].

**Our contributions.** In this work, we provide a principled methodology to measure fairness while accounting for uncertainty and group size. Specifically, we propose a theoretically grounded fairness measure based on hypothesis testing, and we introduce two tests to detect fairness violations. The first test relies on the asymptotic theoretical behavior of statistical parity. In particular, we demonstrate the asymptotic normality of statistical parity, which allows the construction of confidence intervals and tests for fairness violations. Secondly, we propose a test based on Bayesian inference, which is particularly useful when intersectional groups are very small and asymptotic assumptions do not hold. We further provide empirical evidence showing that the Bayesian estimator converges to the theoretical asymptotic behavior. Additionally, we highlight the pitfalls of current fairness measures, especially in the context of intersectional groups. We show that existing metrics can incorrectly detect unfairness in cases where no unfairness exists, and conversely, fail to detect statistically significant discriminatory behavior when it is present.

**How our work advances beyond prior research.** Kearns et al. [33] account for the size of intersectional groups by scaling fairness metrics – such as statistical parity – by multiplying the metrics by the probability of belonging to a subgroup. However, to determine whether a group is treated unfairly, one still needs to rely on a predefined threshold. Kim et al. [34] use Bayesian models to augment labeled data with unlabeled data, producing more accurate and lower-variance estimates. Unlike our work, they focus on unlabeled data and do not provide asymptotic results or hypothesis tests. Foulds et al. [22, 23] employ a Dirichlet-Multinomial model distribution; however, unlike us, they still rely on thresholds to detect which group is treated unfairly and do not provide hypothesis tests for fairness violation. Besse et al. [6, 7] compute the asymptotic distributions of the ratios of certain fairness metrics. We generalize their results to a broader class of fairness metrics and, in contrast to their approach, we also address and connect our findings to the case of small intersectional groups where the large-sample assumption does not hold.

A more exhaustive coverage of related literature is presented in Appendix A.

## 2 Problem Definition

Our objective is to move beyond the arbitrary nature of hand-picked thresholds by introducing a rigorous, uncertainty-aware test for detecting group-level discrimination. To set the stage, we first formalize our notation, then assess existing threshold-based criteria, and finally, we present and motivate our alternative approach rooted in significance testing.

**Notation and preliminary definitions.** Let $\mathcal{X}$ denote the input space and $\mathcal{Y} = \{0, 1\}$ the binary label space. For each individual $x \in \mathcal{X}$, let $y \in \mathcal{Y}$ represent the *ground truth label*, and let $f(x) \in \mathcal{Y}$ denote the predicted label given by a deterministic classifier $f : \mathcal{X} \to \mathcal{Y}$. A prediction $f(x) = 1$ corresponds to a favorable decision. Protected attributes are encoded by a space of $p$ discrete values $\mathbb{S} = \mathbb{S}_1 \times \cdots \times \mathbb{S}_p$. Each $S \in \mathbb{S}$ represents the description of an *intersectional subgroup* and $S(\mathcal{X})$ the population of the subgroup. We also denote $\bar{S}$ for the complement of $S$ in $\mathbb{S}$.

**Definition 1** (Statistical parity, or SP). *For any subgroup $S$ and classifier $f$ we measure disparate positive-decision rates via*

$$\mathrm{SP}(S) := \mathbb{P}(f(x) = 1 \mid x \in S(\mathcal{X})) - \mathbb{P}(f(x) = 1 \mid x \in \bar{S}(\mathcal{X})) . \tag{1}$$

Perfect statistical parity corresponds to $\mathrm{SP}(S) = 0$.

We note that we use $\bar{S}$ as the reference group, following common practice. Alternatively, one could use the entire population as the reference group; however, the differences are typically minimal, especially when $S$ represents a small intersectional group.

**Definition 2** ($\delta$-Statistical parity, or $\delta$SP). *A predictor $f$ satisfies statistical parity at level $\delta$ when*

$$\forall S \in \mathbb{S} : \quad |\mathrm{SP}(S)| \leq \delta . \tag{2}$$

A threshold $\theta$ is typically chosen (such as the Equal Employment Opportunity Commission "four-fifths rule" [28], where $\theta = 0.2$), and unfairness is identified when $|\mathrm{SP}(S)| > \theta$. However, this approach does not account for sample variability, treating a subgroup of size 10 the same as one with 10,000.

To partially compensate for heterogeneous support, Kearns et al. [33] propose

**Definition 3** ($\gamma$-Statistical parity, or $\gamma$SP). *A predictor $f$ is $\gamma-$Statistical Parity subgroup fair if:*

$$\forall S \in \mathbb{S} : \quad |\mathrm{SP}(S)| \cdot \mathbb{P}(x \in S(\mathcal{X})) \leq \gamma . \tag{3}$$

However, in practice, to identify unfairly treated groups we have to choose a threshold $\theta$ and claim fairness violation when $|\mathrm{SP}(S)| \cdot \mathbb{P}(x \in S(\mathcal{X})) > \theta$. Moreover, as Gohar and Cheng [27] observe: "The second term reweighs the difference by the proportion of the size of each subgroup in relation to the population. Consequently, the unfairness of smaller-sized groups is down-weighted in the final $\gamma-$Statistical Parity estimation. *Thus, it may not adequately protect small subgroups, even if they have high levels of unfairness.*"

**From thresholds to significance.** Both (2) and (3) require the practitioner to fix a global threshold $\theta$ without guidance on how to adapt it to subgroup size or estimation variance. Motivated by the shortcomings above and by recent calls for uncertainty-aware auditing [18, 42, 39], we recast fairness checking as a statistical hypothesis test and propose the following definition:

**Definition 4** (Statistical Parity violation (this work)). *Given a significance level $\alpha \in [0, 1]$ and a test statistic $T$ with null hypothesis $H_0 : \mathrm{SP}(S) = 0$, we say a classifier $f$ violates statistical parity for subgroup $S$ if the null hypothesis $H_0$ is rejected at level $\alpha$.*

Hence fairness is assessed by whether the observed disparity could plausibly arise from sampling noise. Crucially, the decision threshold is no longer arbitrary: it is determined by the chosen test and $\alpha$, automatically adjusting to the subgroup's sample size.

## 3 A Statistical Testing Framework for Size-Adaptive Fairness Assessment

In this section we operationalise our guiding idea: *replacing ad-hoc thresholds with statistically principled tests for group–level discrimination that remain valid even when intersectional subgroups are small.*

We do so by proposing two tests to detect statistical parity violations: (i) a large-sample **Wald test** that leverages an analytic asymptotic variance, and (ii) a **Bayesian small-sample test** that yields finite-sample credible intervals via Monte-Carlo draws from a Dirichlet–multinomial posterior. Both deliver a $p$-value (or posterior tail-probability) that can be compared with a user-selected significance level $\alpha$, automatically adapting to subgroup size and sampling noise.

This unified framework yields *size-adaptive, uncertainty-aware* fairness conclusions, removing the need for arbitrary global thresholds while retaining statistical guarantees for both common and rare intersectional groups.

## 3.1 Large-sample test: asymptotic normality of $\mathrm{SP}_n(S)$

For an intersectional group $s$ let us define the following probabilities:

$$p_{0,S} = \mathbb{P}(f(x) = 0, x \in S(\mathcal{X})),\; p_{1,S} = \mathbb{P}(f(x) = 1, x \in S(\mathcal{X})),$$
$$p_{0,\bar{S}} = \mathbb{P}(f(x) = 0, x \in \bar{S}(\mathcal{X})),\; p_{1,\bar{S}} = \mathbb{P}(f(x) = 1, x \in \bar{S}(\mathcal{X})),$$

Furthermore, let us define the marginal $p_S = \mathbb{P}(x \in S(\mathcal{X}))$ and $p_{\bar{S}} = \mathbb{P}(x \in \bar{S}(\mathcal{X}))$.

For an i.i.d. test sample $(x_i)_{i=1}^n$ we consider the following *plug-in estimator* of statistical parity:

$$\mathrm{SP}_n(S) := \frac{\sum_{i=1}^n \mathbb{1}_{f(x_i)=1, x_i \in S(\mathcal{X})}}{\sum_{i=1}^n \mathbb{1}_{x_i \in S(\mathcal{X})}} - \frac{\sum_{i=1}^n \mathbb{1}_{f(x_i)=1, x_i \in \bar{S}(\mathcal{X})}}{\sum_{i=1}^n \mathbb{1}_{x_i \in \bar{S}(\mathcal{X})}},$$

where $\mathbb{1}$ denotes the indicator function. Furthermore, let $V = \left( \frac{p_{0,S}}{(p_S)^2}, \frac{-p_{1,S}}{(p_S)^2}, \frac{-p_{0,\bar{S}}}{(p_{\bar{S}})^2}, \frac{p_{1,\bar{S}}}{(p_{\bar{S}})^2} \right)$ and

$$\Sigma_4 = \begin{pmatrix} p_{1,S}(1-p_{1,S}) & -p_{1,S}p_{0,S} & -p_{1,S}p_{1,\bar{S}} & -p_{1,S}p_{0,\bar{S}} \\ -p_{0,S}p_{1,S} & p_{0,S}(1-p_{0,S}) & -p_{0,S}p_{1,\bar{S}} & -p_{0,S}p_{0,\bar{S}} \\ -p_{1,\bar{S}}p_{1,S} & -p_{1,\bar{S}}p_{0,S} & p_{1,\bar{S}}(1-p_{1,\bar{S}}) & -p_{1,\bar{S}}p_{0,\bar{S}} \\ -p_{0,\bar{S}}p_{1,S} & -p_{0,\bar{S}}p_{0,S} & -p_{0,\bar{S}}p_{1,\bar{S}} & p_{0,\bar{S}}(1-p_{0,\bar{S}}) \end{pmatrix}. \tag{4}$$

The following result (proof in Appendix B) provides the asymptotic distribution of the statistical parity.

**Theorem 1** (Central Limit Theorem for Statistical Parity). *Let $\sigma(S) = \sqrt{V^\top \Sigma_4 V}$, where $V$ and $\Sigma_4$ are defined above. Then*

$$\frac{\sqrt{n}}{\sigma(S)} \left( \mathrm{SP}_n(S) - \mathrm{SP}(S) \right) \xrightarrow{d} N(0,1), \quad \textit{as } n \to \infty,$$

*where $\xrightarrow{d}$ denotes convergence in distribution and $N(0,1)$ indicates the standard normal distribution.*

From Theorem 1 we obtain the $(1 - \alpha)$ two-sided Wald confidence interval:

$$\left[ \mathrm{SP}_n(S) \pm \frac{\sigma(S)}{\sqrt{n}} \Phi^{-1}\left(1 - \frac{\alpha}{2}\right) \right],$$

and the corresponding $p$-value $p = 2\left(1 - \Phi(|\sqrt{n}, \mathrm{SP}_n(S)/\sigma(S)|)\right)$, where $\Phi(\cdot)$ is the cumulative distribution function of the standard normal distribution and $\Phi^{-1}(\cdot)$ the quantile. We reject the null "no disparity" whenever $p < \alpha$. Because $\sigma(S)$ depends on *population* probabilities, we estimate it by plugging empirical counts into Equation (4).

## 3.2 Bayesian inference for small samples

Theorem 1 describes how confidence intervals for $\mathrm{SP}(S)$ can be computed with large samples. In the particular case of small intersectional groups this assumption is, in general, not met and the theoretical asymptotic estimations of $\mathrm{SP}(S)$ and $\Sigma_4$ might not be reliable. We thus move our attention to deriving credible intervals via Bayesian inference.

**Prior, likelihood and posterior.** We consider the probability space over $p_{0,S}, p_{1,S}, p_{0,\bar{S}}$, and $p_{1,\bar{S}}$. Since the $p_{i,j}$, $i \in \{0,1\}, j \in \{S, \bar{S}\}$ are mutually exclusive, we can assume that observations are drawn from a categorical distribution. Hence, given $n$ trials, let $n_{i,j}$, $i \in \{0,1\}, j \in \{S, \bar{S}\}$ denote the counts for each outcome, with $\sum_{i,j} n_{i,j} = n$. $n_{i,j}$ follows a multinomial distribution: $(n_{0,S}, n_{0,\bar{S}}, n_{1,S}, n_{1,\bar{S}}) \sim \mathrm{Multinomial}(n, p_{0,S}, p_{0,\bar{S}}, p_{1,S}, p_{1,\bar{S}})$. One can consider a Dirichlet prior over $p_{i,j}$: $(p_{0,S}, p_{0,\bar{S}}, p_{1,S}, p_{1,\bar{S}}) \sim \mathrm{Dirichlet}(n, \alpha_{0,S}, \alpha_{0,\bar{S}}, \alpha_{1,S}, \alpha_{1,\bar{S}})$. By conjugacy, the posterior distribution remains Dirichlet with updated parameters:

$$(p_{0,S}, p_{0,\bar{S}}, p_{1,S}, p_{1,\bar{S}}) \mid \mathrm{data} \sim \mathrm{Dirichlet}(n, \alpha_{0,S}+n_{0,S}, \alpha_{0,\bar{S}}+n_{0,\bar{S}}, \alpha_{1,S}+n_{1,S}, \alpha_{1,\bar{S}}+n_{1,\bar{S}}) \tag{5}$$

**Posterior draws and credible interval.** We are now interested in the posterior estimate of $\text{SP}(S)$. Since there is not known closed-form expression for the posterior distribution of $\text{SP}(S)$, we can approximate it using Monte Carlo sampling.

The procedure is as follows. Sample $K$ times from the posterior, let the $k$-th sample be: $(p_{0,S}^{(k)}, p_{0,\bar{S}}^{(k)}, p_{1,S}^{(k)}, p_{1,\bar{S}}^{(k)})$, for $k = 1, \ldots, K$, and let the $k$-th estimate of $\text{SP}(S)$ be:

$$\text{SP}^{(k)}(S) := \frac{p_{1,S}^{(k)}}{p_{0,S}^{(k)} + p_{1,S}^{(k)}} - \frac{p_{1,\bar{S}}^{(k)}}{p_{0,\bar{S}}^{(k)} + p_{1,\bar{S}}^{(k)}} \ .$$

Then:

- An unbiased estimate of the posterior mean of $\text{SP}(S)$ is given by:

$$\mathbb{E}[\text{SP}(S) \mid \text{data}] = \frac{1}{K} \sum_{k=1}^{K} \text{SP}^{(k)}(S)$$

- The $(1 - \alpha)$ credible interval is given by:

$$\text{CI}_{1-\alpha}(S) \ = \ \left[ \widehat{Q}_{\frac{\alpha}{2}}\big(\{\text{SP}^{(k)}(S)\}_{k=1}^{K}\big), \ \widehat{Q}_{1-\frac{\alpha}{2}}\big(\{\text{SP}^{(k)}(S)\}_{k=1}^{K}\big) \right] \ ,$$

where the empirical-quantile at level $u$ is defined by:

$$\widehat{Q}_u\big(\{\text{SP}^{(k)}(S)\}_{k=1}^{K}\big) \ = \ \inf\Big\{ t : \frac{1}{K} \sum_{k=1}^{K} \mathbb{1}\{\text{SP}^{(k)}(S) \leq t\} \ \geq \ u \Big\}.$$

The interval reflects posterior uncertainty over the statistical parity difference given the observed data and prior. We can, hence, perform a two-sided hypothesis test for the null $H_0 : \text{SP}(S) = 0$ from the credible interval. Specifically, we reject the null hypothesis at level $\alpha$ if the value $0$ is not contained in the $(1 - \alpha)$ credible interval.

**Choice of prior.** We adopt a weakly informative symmetric (flat) Dirichlet prior, i.e. with concentration parameters $(1, 1, 1, 1)$, by default; domain knowledge (e.g. historical results), when available, can be injected via the prior parameters. We explore the possibility of incorporating additional information into the prior in Appendix C.7.

**Statistical Fairness Testing Framework.** Algorithm 1 summarizes our rigorous statistical framework for fairness assessment, which is specifically designed to handle varying subgroup sizes in intersectional settings. It integrates large-sample hypothesis testing with Bayesian inference, ensuring that fairness evaluations remain reliable even when data availability differs across subgroups. It dynamically adjusts significance thresholds, thus accounting for statistical uncertainty and preventing misleading conclusions about bias.

### 3.3 Resolution limits for statistical parity fairness violation

Auditing rare intersectional subgroups naturally raises the question: how small is too small? Even with uncertainty-aware tests, a subgroup composed of only a few samples may fail to provide statistically conclusive evidence of bias, regardless of how pronounced the observed disparity might be. Our framework addresses this challenge by computing, for each overall negative prediction probability $\mathbb{P}(f(x) = 0)$ and protected group size $n_S = n_{0,S} + n_{1,S}$, the minimum fraction of negative outcomes in $S$ required to reject the null hypothesis $H_0 : \text{SP}(S) = 0$ at a specified significance level $\alpha$.

Figure 1 visualizes this boundary when evaluating the hypothesis that a group is being *disadvantaged* (the dual figure showing the boundary for deciding if a group is being *advantaged*, is reported in Figure 5 of the Appendix). The plot has on the $y$-axis the fraction $p_{0,S}$ of 0s predicted in the given group, while on the $x$-axis the size $n_S$ of the group. The plot reports 9 different lines corresponding to the fraction of 0s predicted in the overall population. The figure uses $\alpha = 0.05$.

**Algorithm 1** Size-Adaptive Fairness Testing (SAFT)

1: **Input:** Subgroup $S$, significance level $\alpha$, minimum support $\tilde{n}$ (we use $\tilde{n}=30$)
2: **Output:** Fairness decision (Reject $H_0$ or Fail to Reject $H_0$)
3: Compute $n_{i,j}$, $i \in \{0,1\}$, $j \in \{S, \bar{S}\}$
4: **if** $\min\{n_{0,S}, n_{1,S}, n_{0,\bar{S}}, n_{1,\bar{S}}\} \geq \tilde{n}$ **then**
5:      apply the Wald test of §3.1 {large-sample regime}
6: **else**
7:      use the Bayesian test of §3.2 {small-sample regime}
8: **end if**
9: Compute statistical parity measure $\mathrm{SP}(S)$
10: Report point estimate $\mathrm{SP}_n(S)$ or posterior mean
11: Report $(1-\alpha)$ confidence/credible interval
12: Report $p$-value or posterior tail probability
13: **if** $p < \alpha$ **then**
14:      Reject $H_0$ {Fairness violation detected}
15: **else**
16:      Fail to reject $H_0$ {No statistically significant violation}
17: **end if**

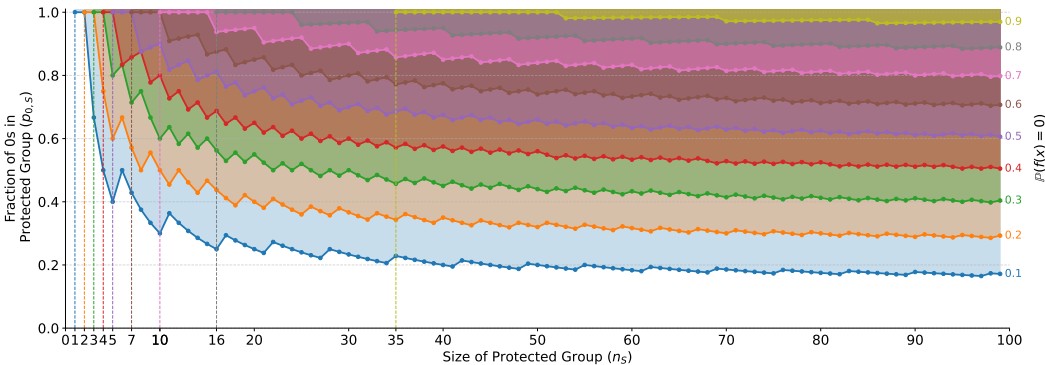

Figure 1: Resolution limits for Statistical Parity violations under varying global negative rates $\mathbb{P}(f(x) = 0)$, when detecting *disadvantaged* groups (the dual figure showing the boundary for deciding if a group is being *advantaged*, is reported in Figure 5 of the Appendix). Each curve traces the minimal fraction of negative outcomes needed to reject $H_0$: $\mathrm{SP}(S) = 0$ at $\alpha = 0.05$ as a function of the group size $n_s$. To the left of each vertical bar is the "no-power" zone, where subgroups are too small to detect discrimination, regardless of the observed disparity. The shaded region above each curve is the "discrimination zone", where the subgroup's negative rate is enough to establish a statistically significant parity violation.

A pair $(n_S, p_{0,S})$ needs to be *above the line* to indicate significant disparity. For instance, the point $(10, 0.6)$ on the green line ($\mathbb{P}(f(x) = 0) = 0.3$) indicates that, when in the overall population there are 30% of 0s predicted, a demographic subgroup of size 10 needs to have at least 6 individuals predicted 0, in order to have a negative discrimination case. When in the overall population there are 40% of 0s predicted (red line), a group of size 10 requires 8 negative cases to have unfairness. When instead the prediction in the overall population is balanced (purple line, $\mathbb{P}(f(x) = 0) = 0.5$) a subgroup of size 10 requires at least 9 negative cases to have a legitimate discrimination complaint.

We can observe that, as the global negative rate $\mathbb{P}(f(x) = 0)$ increases, the required subgroup size $n_s$ grows steeply to evaluate *disadvantaged* groups. For example, when 90% of the population is predicted negative, a group needs to be composed of at least 35 individuals, all predicted negative, to have any statistically valid unfairness, highlighting the difficulty of auditing under severe class imbalance.

By charting these boundaries, our framework highlights the resolution limits of intersectionality, while providing practitioners a tool to determine exactly how much data each intersectional slice needs to be audit-worthy, preventing spurious signalling of fairness violations on small subgroups.

## 3.4 Generalisation to other fairness measures

Analogous results can be derived for other fairness metrics. Indeed, it holds the general form of the theorem (we defer the proof to Appendix B):

**Theorem 2.** *Let $\boldsymbol{p} = (p_1, \ldots, p_q)$ a set of probabilities of $q$ disjoint events, with $\sum_i^q p_i = 1$; let $C = (C_1, \ldots, C_q) \sim$ Categorical$(\boldsymbol{p})$ be the related categorical distribution, and let $C^1, \ldots, C^n$ be $n$ i.i.d. realizations of $C$. Let $\phi$ be a continuously differentiable (in an neighborhood of $\mathbb{E}C$) function $\phi : \mathbb{R}^q \to \mathbb{R}$. Then*

$$\frac{\sqrt{n}}{\sigma} \left( \phi(\frac{1}{n} \sum_{i=1}^n C^i) - \phi(\mathbb{E}C) \right) \xrightarrow{d} N(0,1), \quad as\ n \to \infty,$$

*where $\sigma = \sqrt{V^\top \Sigma V}$, $V = \nabla(\phi(\mathbb{E}C))$ is the gradient of $\phi$, and $\Sigma = [\mathrm{diag}(\boldsymbol{p}) - \boldsymbol{p}\boldsymbol{p}^T]$ is the covariance matrix of $C$, where $\mathrm{diag}(\boldsymbol{p})$ indicates a diagonal matrix with entries $p_i$.*

For example, considering **Equal Opportunity**:

$$\mathrm{EO}(S) \coloneqq \mathbb{P}(f(x) = 1 \mid y = 1,\ x \in S(\mathcal{X})) - \mathbb{P}(f(x) = 1 \mid y = 1,\ x \in \bar{S}(\mathcal{X})),$$

Theorem 2 can be applied by choosing $p_1 = \mathbb{P}(f(x) = 0, x \in S(\mathcal{X}) | y = 1)$, $p_2 = \mathbb{P}(f(x) = 1, x \in S(\mathcal{X}) | y = 1)$, $p_3 = \mathbb{P}(f(x) = 0, x \in \bar{S}(\mathcal{X}) | y = 1)$, $p_4 = \mathbb{P}(f(x) = 1, x \in \bar{S}(\mathcal{X}) | y = 1)$, and $\phi(x_1, x_2, x_3, x_4) = \frac{x_2}{x_1 + x_2} - \frac{x_4}{x_3 + x_4}$.

From Theorem 2, one can also obtain equivalent results to [6, 7] (but with a simpler computation of the covariance matrix). Indeed, for example, the asymptotic behaviour of the **Disparate Impact** assessment:

$$\mathrm{DI}(S) \coloneqq \frac{\mathbb{P}(f(x) = 1 \mid x \in S(\mathcal{X}))}{\mathbb{P}(f(x) = 1 \mid x \in \bar{S}(\mathcal{X}))},$$

can be obtained by choosing $p_1 = \mathbb{P}(f(x) = 0, x \in S(\mathcal{X}))$, $p_2 = \mathbb{P}(f(x) = 1, x \in S(\mathcal{X}))$, $p_3 = \mathbb{P}(f(x) = 0, x \in \bar{S}(\mathcal{X}))$, $p_4 = \mathbb{P}(f(x) = 1, x \in \bar{S}(\mathcal{X}))$, and $\phi(x_1, x_2, x_3, x_4) = \frac{x_2}{x_1 + x_2} \cdot \frac{x_3 + x_4}{x_4}$.

The results for the Bayesian test can be extended under analogous conditions. Indeed, one can consider a general probability space over $\mathbf{p} = (p_1, \ldots, p_q)$ of $q$ disjoint events. Consider $n$ trials and indicate with $(n_1, \ldots, n_q)$ the counts for each outcome. The likelihood is again Multinomial$(n, \mathbf{p})$, and with a Dirichlet prior over $\mathbf{p}$ with concentration parameters $(\alpha_1, \ldots, \alpha_q)$, by conjugacy we obtain a posterior distribution distributed as a Dirichlet$(n, \alpha_1 + n_1, \ldots, \alpha_q + n_q)$. One can then consider a fairness metric $\phi : \mathbb{R}^q \to \mathbb{R}$ and via Monte Carlo sampling can construct a $(1 - \alpha)$ credible interval and hypothesis test for $\phi$, analogously to the procedure used for $\mathrm{SP}(S)$.

In general, our framework can model all those fairness metrics based on the confusion matrix and related conditional probabilities.

## 4 Experiments

In this section, we empirically validate our size-adaptive testing framework on two canonical fairness benchmarks under increasingly fine-grained intersectional slices.

### 4.1 Datasets and Models

We selected two standard datasets [1] used in fairness benchmarks: the Adult Income dataset [5] and the COMPAS recidivism dataset [2]. We include additional experiments on two other datasets in the Appendix. We apply standard data preprocessing and train an XGBoost classifier $g : \mathcal{X} \to \mathcal{Y}$ (training details are in Appendix C.2). To examine intersectional effects, we first audit non-intersectional subgroups (*race*, *age* and *sex*) and then all two-way and three-way intersections of these protected attributes.

---

[1]COMPAS from propublica (compas-scores-two-years.csv); Adult from fairlearn.org, originally from UCI.

**COMPAS**  The COMPAS dataset is widely used in the study of algorithmic fairness and risk assessment in the criminal justice system. It is often cited in fairness literature due to observed disparities in prediction outcomes across racial groups, most notably between Black and White defendants [2]. It includes information about criminal defendants, such as the number of prior convictions and charge degree. The protected variables are $race$ ("African-American", "Asian", "Caucasian", "Hispanic", "Native American", "Other"), $age$ ("Under 25", "25-45", "Over 45"), and $sex$ ("Female", "Male"). The binary target variable indicates whether a defendant is predicted to reoffened within two years ($y = 0$) or not ($y = 1$).

**Adult**  The Adult Income dataset is based on census data, which includes 14 categorical and numerical features (e.g. education, marital status, occupation, etc.). The protected variables are $race$ ("Amer-Indian-Eskimo", "Asian-Pacific-Islander", "Black", "Other", "White"), $age$ ("Under 18", "18-40", "40-65", "Over 65"), and $sex$ ("Female", "Male"). The binary target is whether the individual annual income exceeds $50k$ per year ($y = 1$) or not ($y = 0$).

**Reproducibility**  All experiments were run on a server equipped with an Intel Xeon Gold 6312U CPU and 256 GB of RAM. Our full codebase—including data preprocessing, model training, and auditing notebooks—can be found at `https://github.com/alanturin-g/SAFT`.

### 4.2  Comparison with point-wise estimations

We consider both COMPAS and Adult datasets over 20 random 2:1 train-test splits, followed by model training and fairness auditing each subgroup on every split. In each case, we compare our approach against the fixed-threshold $\delta$SP criterion ($\theta = \pm.1$, as, e.g., in [1]), to highlight how a threshold-based audit, while common, fails to account for sampling variability.

Figure 2 illustrates a selection of intersectional subgroups (one per panel) from the COMPAS dataset. In each panel, we specify the minimum ($n_{min}$) and maximum ($n_{max}$) size of the protected group across different train-test splits. The gray band marks the no-detection region under a fixed-threshold approach. For each train-test split we compute the point-wise SP estimation, the Bayesian (blue) and the asymptotic (red) credible/confidence intervals. In panel (a) we observe a scenario where $\delta$SP would detect a violation, since all point estimates are outside the gray band, and our approach agrees, since we can reject $H_0$. In panel (b), with a smaller intersectional group, $\delta$SP would detect fairness violations, but our approach shows that, due to the wide confidence interval, we cannot reject $H_0$ and detect a violation. It is worth observing that in this scenario, due to the small size of the protected group, the asymptotic behaviour would not be reliable; our approach SAFT would indeed rely on the Bayesian inference instead. In panel (c), we show how, for small groups, the train-test split can have a strong impact: $\delta$SP would detect fairness violations in roughly half the cases, while our answer is consistent across all splits. A similar pattern, but with a fairness violation consistently detected on our side, is shown in panel (d) for a bigger group.

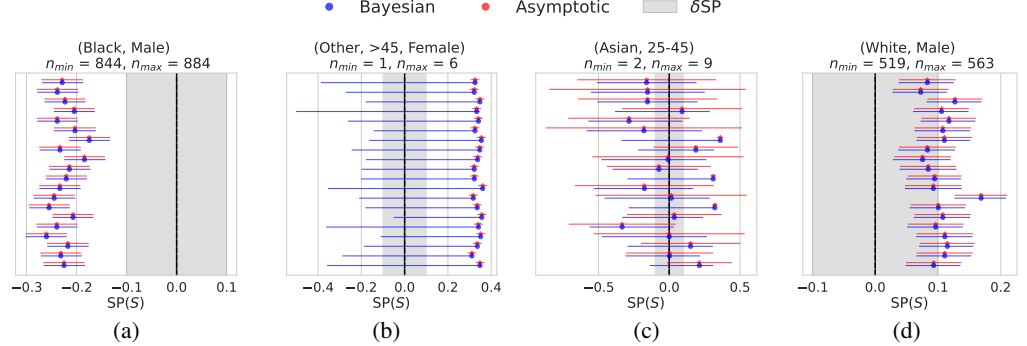

Figure 2: Point-wise estimation versus confidence intervals, COMPAS dataset.

Figure 3 shows analogous results for the Adult dataset. In panel (a) we show a cross-split, cross-approach agreement on a fairness violation detection. In panel (b) we have a disagreement, as $\delta$SP would detect fairness violations, but our approach cannot reject $H_0$. In panel (c) we have the opposite disagreement: $\delta$SP would not detect fairness violations, but we show that with such big group $H_0$ can

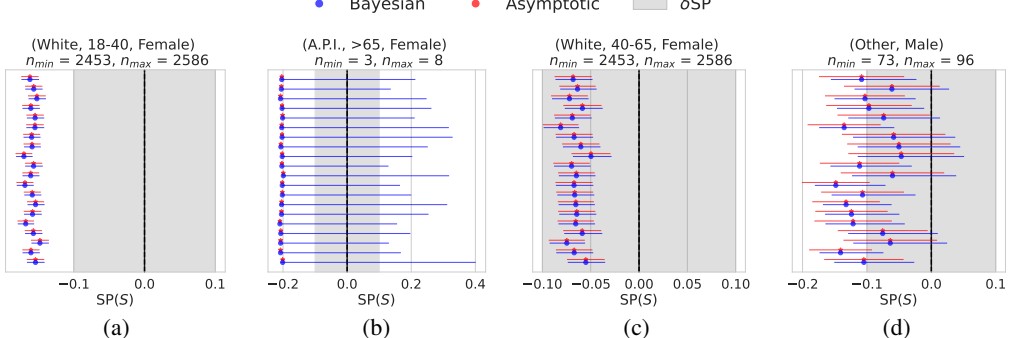

Figure 3: Point-wise estimation versus confidence intervals, Adult dataset.

be rejected, and a fairness violation warning raised. In panel (c) we show a scenario where the train-test split impacts both $\delta$SP and our approach. In general, these results show how a fixed-threshold approach cannot accommodate groups of different size even within the same dataset-model scenario, and can be sensitive to different train-test split. Conversely, our approach takes into account the group size and is generally more robust regarding train-test splits.

### 4.3 Comparison with $\gamma$SP

We again consider all intersectional groups (across all 20 train-test splits) on COMPAS and Adult to compare the decision from our framework with those from a threshold-based $\gamma$SP approach. We plot the size of the protected group against its $\gamma$SP score computed according to Equation 3. We also compute the fairness violation with our statistical testing approach SAFT (Algorithm 1), and use the results as color-code in the scatterplots: red for fairness violation detection, blue for no violation. Figure 4 (COMPAS on the left, Adult on the right) shows that no single threshold (horizontal line) on $\gamma$SP can cleanly separate true violations from non-violations. These plots further support the need for a size-adaptive hypothesis-testing approach.

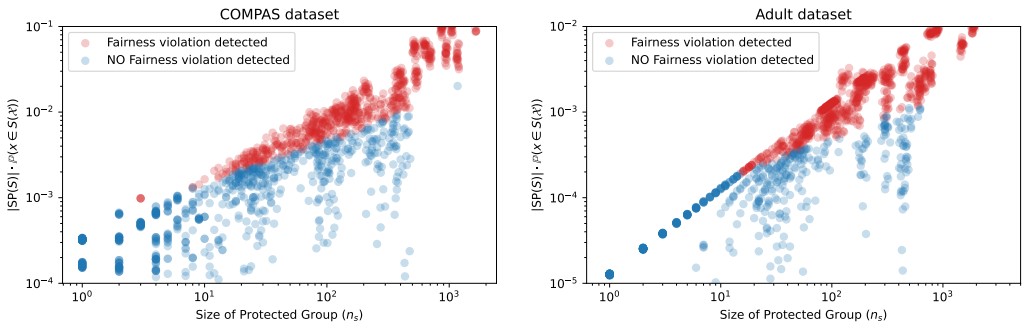

Figure 4: Protected groups size, $\gamma$SP scores, and interval-based fairness violations.

## 5   Conclusions

In this work, we have shown how fairness auditing can move beyond brittle point estimates and arbitrary thresholds. By providing an analytic Central-Limit result for the statistical parity difference and coupling it with a Bayesian Dirichlet-multinomial estimator, our framework conveys size-adaptive confidence and credible intervals that remain valid from the most common to the rarest intersectional subgroups.

**Limitations.** Our Bayesian framework inherently requires the specification of a prior distribution, which encodes assumptions about the parameters before observing the data. Even when employing weakly informative (flat) priors, one is still making a modelling choice. However, this is standard within the Bayesian framework and should be understood as a natural consequence of its formulation.

Moreover, while we discuss the generalization to other fairness measures, our focus is on statistical parity (and on equal opportunity in Appendix C.9) and leaves open empirical validation for additional group fairness definitions.

**Future work.** Looking ahead, we intend to extend our methodology to consider tests for individual fairness [17, 25, 21] and counterfactual fairness [35], which evaluate whether the model behaves fairly for individuals and under hypothetical changes to sensitive attributes, and represent an additional classes of fairness metrics, different from the group fairness metrics considered in this work.

**Broader Impact.** Our testing framework strengthens fairness audits by providing transparent, size-adaptive confidence intervals, which can reduce both false alarms and overlooked harms, especially for under-represented intersectional groups. At the same time, because fairness admits many metrics and any auditor can choose the one most favourable to their goals, there is a risk of using selective reporting to hide genuine bias. We therefore urge practitioners to apply our methods responsibly: report all tested metrics and provide transparent interpretations of confidence intervals rather than as standalone justifications for deployment.

## Acknowledgments and Disclosure of Funding

This work is conducted as part of the Horizon Europe project PRE-ACT (Prediction of Radiotherapy side effects using explainable AI for patient communication and treatment modification). It is supported by the European Commission through the Horizon Europe Program (Grant Agreement number 101057746), by the Swiss State Secretariat for Education, Research and Innovation (SERI) under contract number 22 00058, and by the UK government (Innovate UK) under application number 10061955.

Furthermore, the authors wish to thank Filippo Ascolani for the fruitful discussions regarding the statistical framework.

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

# Appendices

## A  Background and Related Work

Early research on algorithmic fairness framed the problem as ensuring Statistical Parity across sensitive groups defined by single attributes such as race or gender. Demographic Parity [19], Equalized Odds and Equal Opportunity [29], Predictive Parity [12], and related criteria remain cornerstones of the field and are implemented in popular toolkits (e.g., FAIRLEARN [44]). A large body of work subsequently proposed pre-, in-, and post-processing interventions to satisfy one or more of these metrics [46, 24].

While the above metrics are ordinarily reported as single numbers, they are in fact noisy estimates computed on finite samples. Angwin et al. [2] and the ensuing debate around COMPAS highlighted how small data perturbations can flip an "unfair" judgment. Kearns et al. [33] observed that even with large overall data sets, fairness estimates for minority sub-groups can have extremely high variance, a problem magnified under intersectionality. Recent surveys [7, 8] consolidate evidence that ignoring estimation error leads to both false alarms (flagging random fluctuation as bias) and missed harms (overlooking small but significant disparities in large groups).

Intersectionality [15, 27, 26, 20] demands analysis at the cross-sections of multiple protected attributes (e.g. "Black women over 50"). Empirical studies such as *Gender Shades* [9] revealed accuracy gaps that only appear at such intersections. Yet the number of subgroups grows exponentially with attribute count, leaving most groups sparsely represented. Foulds et al. [23] formalized this "resolution limit" and demonstrated that point estimates become statistically meaningless in the low-data regime. Subsequent work proposed auditing algorithms that search adaptively for large-effect subgroups without exhaustively enumerating all intersections [33, 40], but these methods still deliver binary verdicts on noisy estimates. Complementary to these estimation strategies, Molina and Loiseau [39] provide high-probability *bounds* that connect easily measured *marginal* fairness to otherwise intractable intersectional fairness. Their analysis shows when marginal guarantees suffice and offers computable certificates that scale to many attributes.

A direct frequentist remedy is to report confidence intervals (CIs) around each fairness metric. Bootstrap CIs for Demographic Parity, Equalized Odds, and related metrics are supported in FAIRLEARN

and have been advocated by Besse et al. [6], Del Barrio et al. [16], and Cherian and Candès [11]. In NLP, Ethayarajh [18] introduces *Bernstein-bounded unfairness*, deriving analytic Bernstein bounds that convert a small, partially annotated sample into a tight confidence interval on bias. Wide intervals immediately signal when data are insufficient to draw conclusions, a property exploited in the *FairlyUncertain* benchmark [3]. Nevertheless, Bootstrap CIs are not reliable when sample sizes are extremely small. Additionally, Himmelreich et al. [31] advocate for hypothesis testing for fairness in intersectional settings; however, differently from us they focus on the accuracy computed per group and solely rely on central limit results, without considering alternatives for small sample sizes.

Beyond estimating metrics more precisely, several authors treat disparities in predictive uncertainty themselves as indicators of unfairness. Bhatt et al. [8] decompose aleatoric and epistemic uncertainty across groups; higher epistemic uncertainty for a subgroup suggests under-representation in the training data. Kuzucu et al. [36] leverage uncertainty estimates from deep ensembles and Bayesian neural networks to design uncertainty-aware mitigation strategies.

Existing solutions thus improve fairness assessment along isolated axes–frequentist intervals or Bayesian models–but none provides a unified, scalable framework that (i) quantifies estimation uncertainty, (ii) adapts gracefully to the extreme sparsity of intersectional subgroups, and (iii) remains agnostic to the choice of fairness metric. We address this gap by introducing a statistically principled method that merges Bayesian modeling with concentration-inequality guarantees, yielding valid uncertainty bounds at small subgroup granularity without prohibitive computation.

# B  Proofs

We first prove Theorem 2 which is the general version. Then, we show how Theorem 1 follows consequently.

**Theorem 2.** *Let $\boldsymbol{p} = (p_1, \ldots, p_q)$ a set of probabilities of $q$ disjoint events, with $\sum_i^q p_i = 1$; let $C = (C_1, \ldots, C_q) \sim Categorical(\boldsymbol{p})$ be the related categorical distribution, and let $C^1, \ldots, C^n$ be $n$ i.i.d. realizations of $C$. Let $\phi$ be a continuously differentiable (in an neighbourhood of $\mathbb{E}C$) function $\phi : \mathbb{R}^q \to \mathbb{R}$. Then*

$$\frac{\sqrt{n}}{\sigma} \left( \phi(\frac{1}{n} \sum_{i=1}^n C^i) - \phi(\mathbb{E}C) \right) \xrightarrow{d} N(0, 1) , \quad as \, n \to \infty ,$$

*where $\sigma = \sqrt{V^\top \Sigma V}$, $V = \nabla(\phi(\mathbb{E}C))$ is the gradient of $\phi$, and $\Sigma = [\mathrm{diag}(\boldsymbol{p}) - \boldsymbol{p}\boldsymbol{p}^T]$ is the covariance matrix of $C$, where $\mathrm{diag}(\boldsymbol{p})$ indicates a diagonal matrix with entries $p_i$.*

*Proof of Theorem 2.* Let $C^1, \ldots, C^n$ be i.i.d. samples from the categorical distribution $C \sim$ Categorical($\mathbf{p}$), where each $C^i$ is a one-hot vector in $\mathbb{R}^q$ (i.e., $C^i \in \{e_1, \ldots, e_q\}$ and $\mathbb{E}C^i = \mathbb{E}C = \mathbf{p}$).

Define the sample mean:

$$\bar{C}_n := \frac{1}{n} \sum_{i=1}^n C^i .$$

Then, by the multivariate Central Limit Theorem [37], we have:

$$\sqrt{n}(\bar{C}_n - \mathbf{p}) \xrightarrow{d} \mathcal{N}(0, \Sigma) ,$$

where $\Sigma = \mathrm{diag}(\mathbf{p}) - \mathbf{p}\mathbf{p}^\top$ is the covariance matrix of the categorical variable $C$.

Now, let $\phi : \mathbb{R}^q \to \mathbb{R}$ be a function continuously differentiable in an neighbourhood of $\mathbf{p} = \mathbb{E}C$. Then, by the multivariate Delta Method [37], we have:

$$\sqrt{n} \left( \phi(\bar{C}_n) - \phi(\mathbf{p}) \right) \xrightarrow{d} \mathcal{N}(0, \sigma^2) ,$$

where $\sigma^2 = V^\top \Sigma V$, with $V = \nabla \phi(\mathbb{E}C) = \nabla \phi(\mathbf{p})$, and, hence:

$$\frac{\sqrt{n}}{\sigma} \left( \phi(\bar{C}_n) - \phi(\mathbf{p}) \right) \xrightarrow{d} \mathcal{N}(0, 1) .$$

$\square$

Furthermore, we point out that by Slutsky's Theorem [43] one can use the covariance $\hat{\sigma}$ estimated from the data, instead of the population one.

We now show that the asymptotic normality of the Statistical Parity estimator $\mathrm{SP}_n(S)$ (Theorem 1) follows from Theorem 2 by expressing it as a smooth function of the empirical mean of a categorical random variable.

**Theorem 1** (Central Limit Theorem for Statistical Parity). *Let* $\sigma(S) = \sqrt{V^\top \Sigma_4 V}$, *where* $V = \left( \frac{p_{0,S}}{(p_S)^2}, \frac{-p_{1,S}}{(p_S)^2}, \frac{-p_{0,\bar{S}}}{(p_{\bar{S}})^2}, \frac{p_{1,\bar{S}}}{(p_{\bar{S}})^2} \right)$, *and* $\Sigma_4$ *is defined in Equation 4 of the paper. Then*

$$\frac{\sqrt{n}}{\sigma(S)} \left( \mathrm{SP}_n(S) - \mathrm{SP}(S) \right) \xrightarrow{d} N(0,1) , \quad \text{as } n \to \infty ,$$

*where* $\xrightarrow{d}$ *denotes convergence in distribution and* $N(0,1)$ *indicates the standard normal distribution.*

*Proof of Theorem 1.* Define the following four mutually exclusive and exhaustive events:

$$e_1 \equiv (f(x) = 1, \ x \in S(\mathcal{X})) , \qquad\qquad e_2 \equiv (f(x) = 0, \ x \in S(\mathcal{X})) ,$$
$$e_3 \equiv (f(x) = 1, \ x \in \bar{S}(\mathcal{X})) , \qquad\qquad e_4 \equiv (f(x) = 0, \ x \in \bar{S}(\mathcal{X})) .$$

Let $C \in \mathbb{R}^4$ be a categorical random variable representing a one-hot encoding over these events, with probability vector $\mathbf{p} = (p_{1,S}, p_{0,S}, p_{1,\bar{S}}, p_{0,\bar{S}})$.

Let $C^1, \ldots, C^n$ be i.i.d. realizations of $C$, and define the empirical mean:

$$\hat{p} = \frac{1}{n} \sum_{i=1}^{n} C^i .$$

Then $\hat{p}$ is a consistent estimator of the population vector $\mathbb{E}C = \mathbf{p}$.

Now define the function $\phi : \mathbb{R}^4 \to \mathbb{R}$ by

$$\phi(p_1, p_2, p_3, p_4) = \frac{p_1}{p_1 + p_2} - \frac{p_3}{p_3 + p_4} ,$$

which corresponds to the population Statistical Parity:

$$\mathrm{SP}(S) = \phi(\mathbf{p}) , \quad \mathrm{SP}_n(S) = \phi(\hat{p}) .$$

Note that $\phi$ is continuously differentiable in a neighborhood of $\mathbf{p}$, provided the denominators $p_1 + p_2$ and $p_3 + p_4$ are nonzero, which holds under standard assumptions.

By Theorem 2, we have the convergence:

$$\frac{\sqrt{n}}{\sigma(S)} \left( \phi(\hat{p}) - \phi(\mathbf{p}) \right) \xrightarrow{d} \mathcal{N}(0,1) ,$$

where $\sigma(S) = \sqrt{V^\top \Sigma V}$, $V = \nabla \phi(\mathbf{p})$, and $\Sigma = \mathrm{diag}(\mathbf{p}) - \mathbf{p}\mathbf{p}^\top$ is the covariance matrix of the categorical variable $C$.

Direct computation of the gradient yields:

$$V = \left( \frac{p_{0,S}}{(p_S)^2}, \frac{-p_{1,S}}{(p_S)^2}, \frac{-p_{0,\bar{S}}}{(p_{\bar{S}})^2}, \frac{p_{1,\bar{S}}}{(p_{\bar{S}})^2} \right) ,$$

with $p_S = p_{1,S} + p_{0,S}$ and $p_{\bar{S}} = p_{1,\bar{S}} + p_{0,\bar{S}}$.

The covariance matrix $\Sigma$ for this 4-category categorical variable coincides with the matrix $\Sigma_4$ defined in Equation (4) of the main paper. Hence, the asymptotic variance is exactly

$$(\sigma(S))^2 = V^\top \Sigma_4 V .$$

Thus,

$$\frac{\sqrt{n}}{\sigma(S)} \left( \mathrm{SP}_n(S) - \mathrm{SP}(S) \right) \xrightarrow{d} \mathcal{N}(0,1) ,$$

which proves Theorem 1 as a special case of Theorem 2. $\qquad\qquad\square$

The results for Equal Opportunity and Disparate Impact are obtained analogously, as special cases of Theorem 2.

# C    Additional experiments and discussions

Due to space constraints, the main paper focused on a selected set of datasets and results. Here, we present broader discussions and analysis.

## C.1    Additional datasets

The additional experiments include the German Credit [32] and the Student Performance [13] datasets.

**German**    The German Credit dataset, sourced from a German bank, comprises 1000 loan applicants characterized by twenty attributes (three of which are considered protected), including financial and personal details. Its primary use is in binary classification tasks to predict creditworthiness, with binary target "bad" ($y = 0$) and "good" ($y = 1$). The three protected variables considered are: *Sex* ("Male", "Female"), *Foreign worker* ("Foreign", "Not foreign"), and *Age* ("Under 30","30-40","40-50","Over 50").

**Student**    The Student Performance dataset focuses on student achievement in secondary education at two Portuguese schools. It includes attributes related to student grades, demographics, social factors, and school-related characteristics. The data was gathered for $649$ students through school reports and questionnaires. The binary target variable indicates whether a student's final grade falls within the range of 0 to 10 ($y = 0$) or exceeds 10 ($y = 1$). The protected variables considered are: *age* ("Adult" ($\geq 18$), "Minor" ($< 18$)), *Sex* ("F", "M") for Female and Male students, *Pstatus* ("T","A"), which indicates the parent's cohabitation status, living Together or Apart, and *Address* ("U","R"), which indicates student's home address type, Urban or Rural areas.

## C.2    Training Details

In all experiments, we train an XGBoost classifier [10] on each dataset without hyperparameter tuning, since our focus is on fairness auditing rather than maximizing predictive performance. We perform 20 independent 2:1 stratified train–test splits to account for sampling variability. The following default XGBoost parameters were used for all runs:

- `n_estimators = 100`
- `max_depth = 6`
- `learning_rate = 0.3`
- `subsample = 1.0`
- `colsample_bytree = 1.0`
- `objective = "binary:logistic"`
- `eval_metric = "logloss"`

After training on each split, we evaluate accuracy on the held-out test set. Table 1 reports the mean and standard deviation of test accuracy over the 20 splits for each dataset.

Table 1: Mean accuracy and standard deviation of the XGBoost model over 20 random 2:1 splits.

| Dataset | Mean Accuracy | Std. Deviation |
|---|---|---|
| Adult | 0.8702 | 0.0021 |
| COMPAS | 0.7141 | 0.0094 |
| German | 0.7633 | 0.0229 |
| Student | 0.7681 | 0.0177 |

## C.3    Additional results for Statistical Parity

Figure 5 shows the resolution limits for Statistical Parity violations when detecting when a group is *advantaged*.

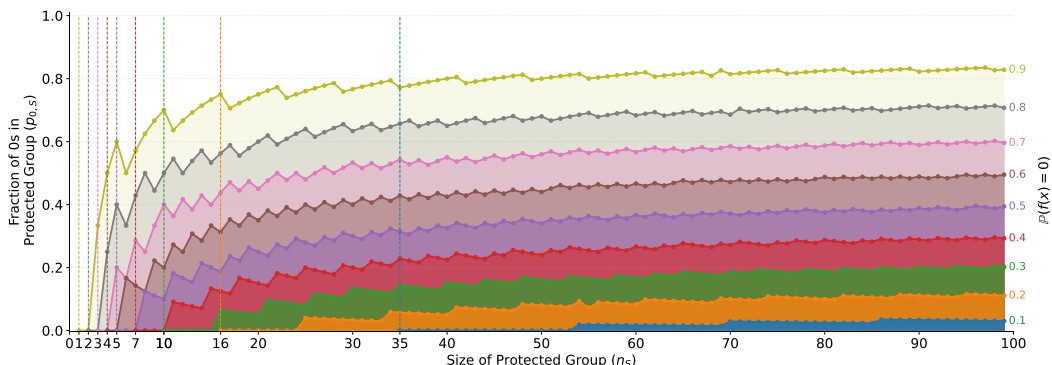

Figure 5: Resolution limits for Statistical Parity violations under varying global negative rates $\mathbb{P}(f(x) = 0)$, when detecting *advantaged* groups (the dual figure showing the boundary for deciding if a group is being *disadvantaged* is reported in Figure 1 in the main paper). Each curve traces the maximal fraction of negative outcomes needed to reject $H_0: \mathrm{SP}(S) = 0$ at $\alpha = 0.05$ as a function of the group size $n_s$. To the left of each vertical bar is the "no-power" zone, where subgroups are too small to detect discrimination, regardless of the observed disparity. The shaded region below each curve is the "discrimination zone", where the subgroup's negative rate is enough to establish a statistically significant parity violation.

In Figures 2 and 3 of the main paper, we showed the results for Statistical Parity for a selection of intersectional subgroups (one per panel) from the COMPAS and Adult datasets. In each panel, we specified the minimum ($n_{min}$) and maximum ($n_{max}$) size of the protected group across different train-test splits. In Figures 6, 7, 8, 9, 10, 11 we show the complete set of intersectional subgroups for the four datasets *Adult*, *COMPAS*, *German* and *Student*. We have omitted panels for empty intersectional groups (('Asian', 'Less than 25', 'Female') and ('N.A.', 'Less than 25', 'Female') in COMPAS, ('A.P.I.', 'Under 18', 'Male') in Adult). For every split, we plot the Bayesian credible interval (blue), and the asymptotic normal confidence interval (red). In certain panels, such as the final one in Figure 10, some lines are absent. This occurs because, in some train-test splits, the intersectional group had no instances in the test fold.

As for the results in the main paper, these findings highlight the limitations of a fixed-threshold approach, which struggles to account for variations in group size within the same dataset-model setup and remains sensitive to different train-test splits. In contrast, our method considers group size, demonstrating greater robustness across train-test variations.

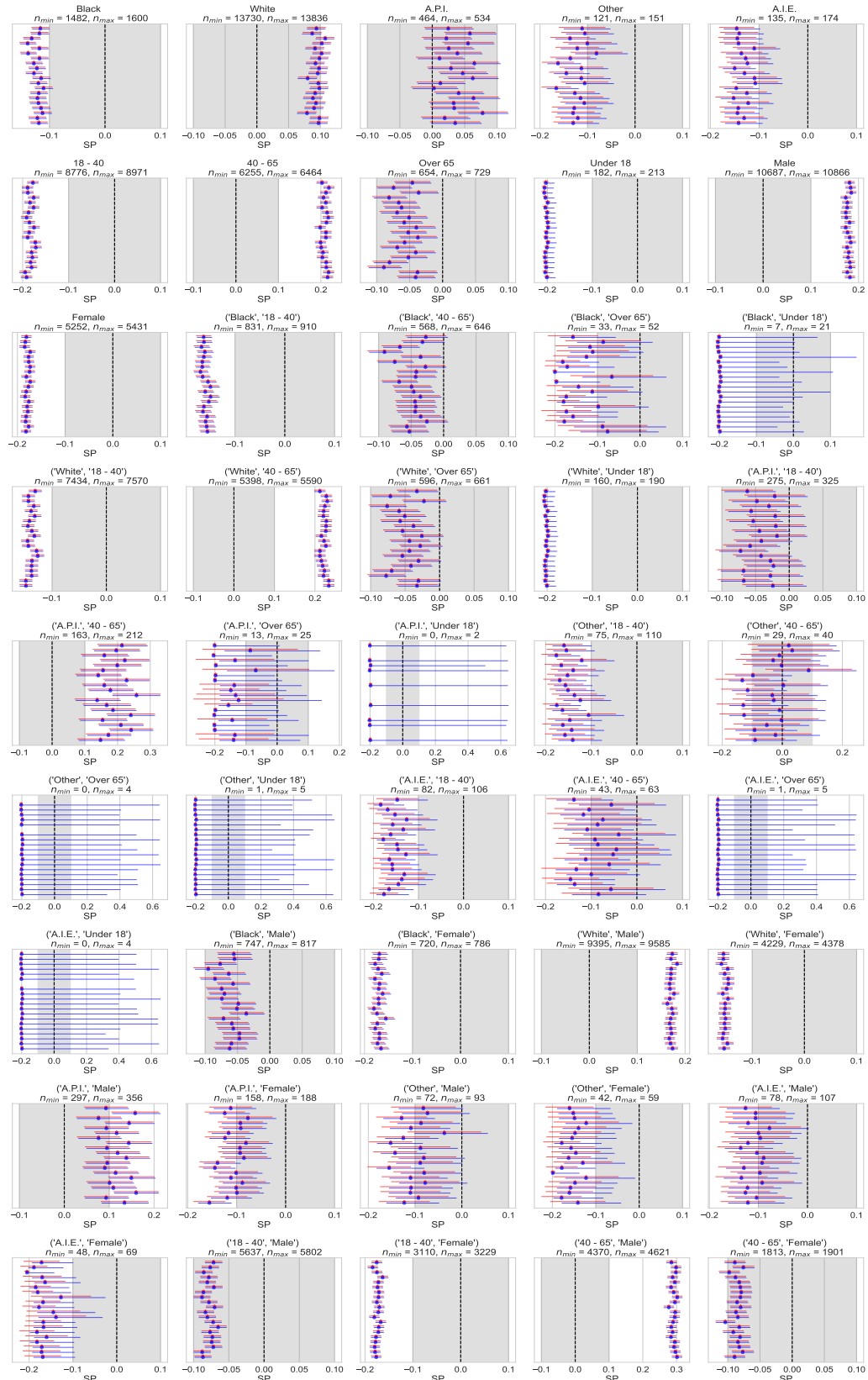

Figure 6: Point-wise estimation versus confidence intervals, Adult dataset.

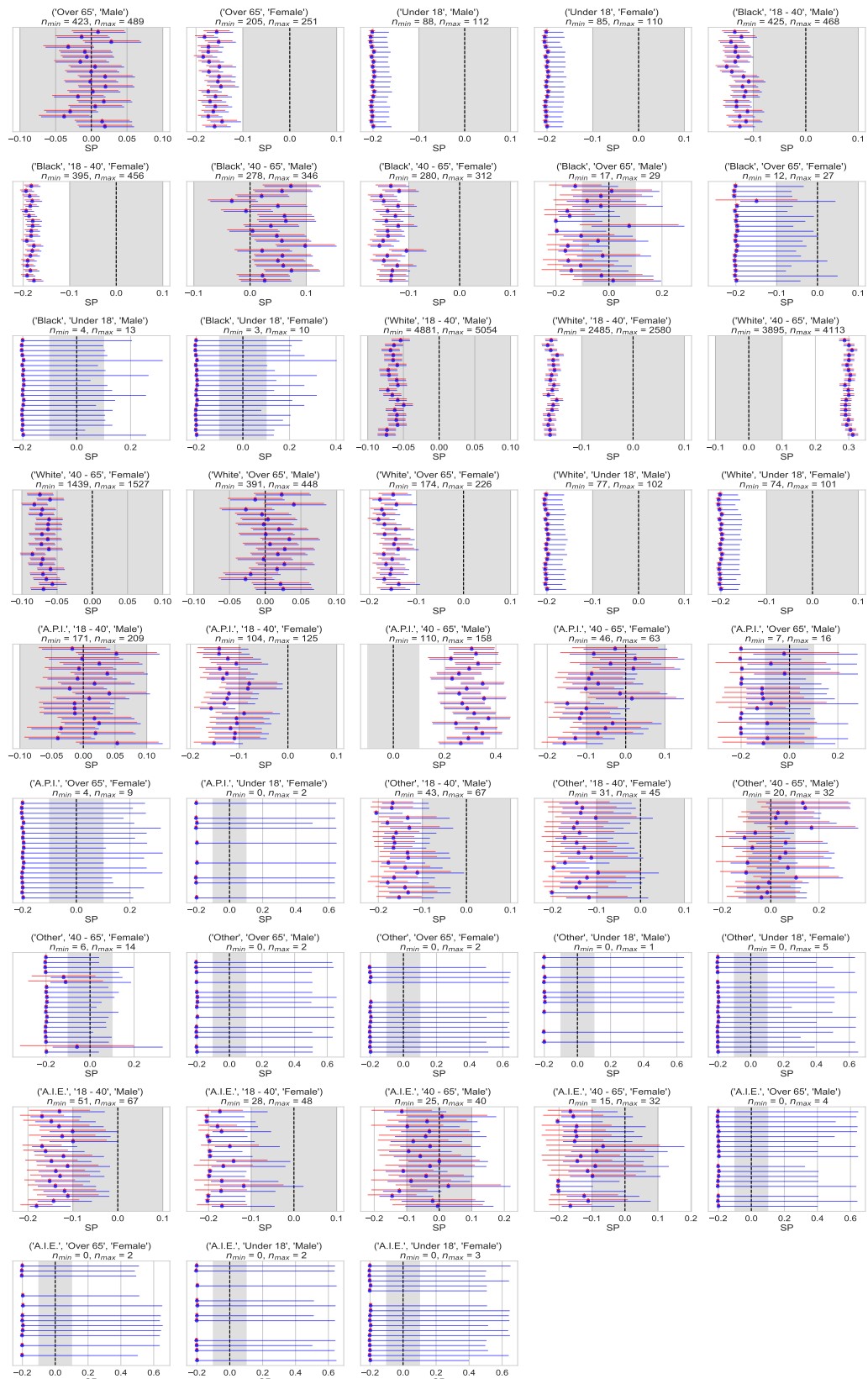

Figure 7: Point-wise estimation versus confidence intervals, Adult dataset (continued).

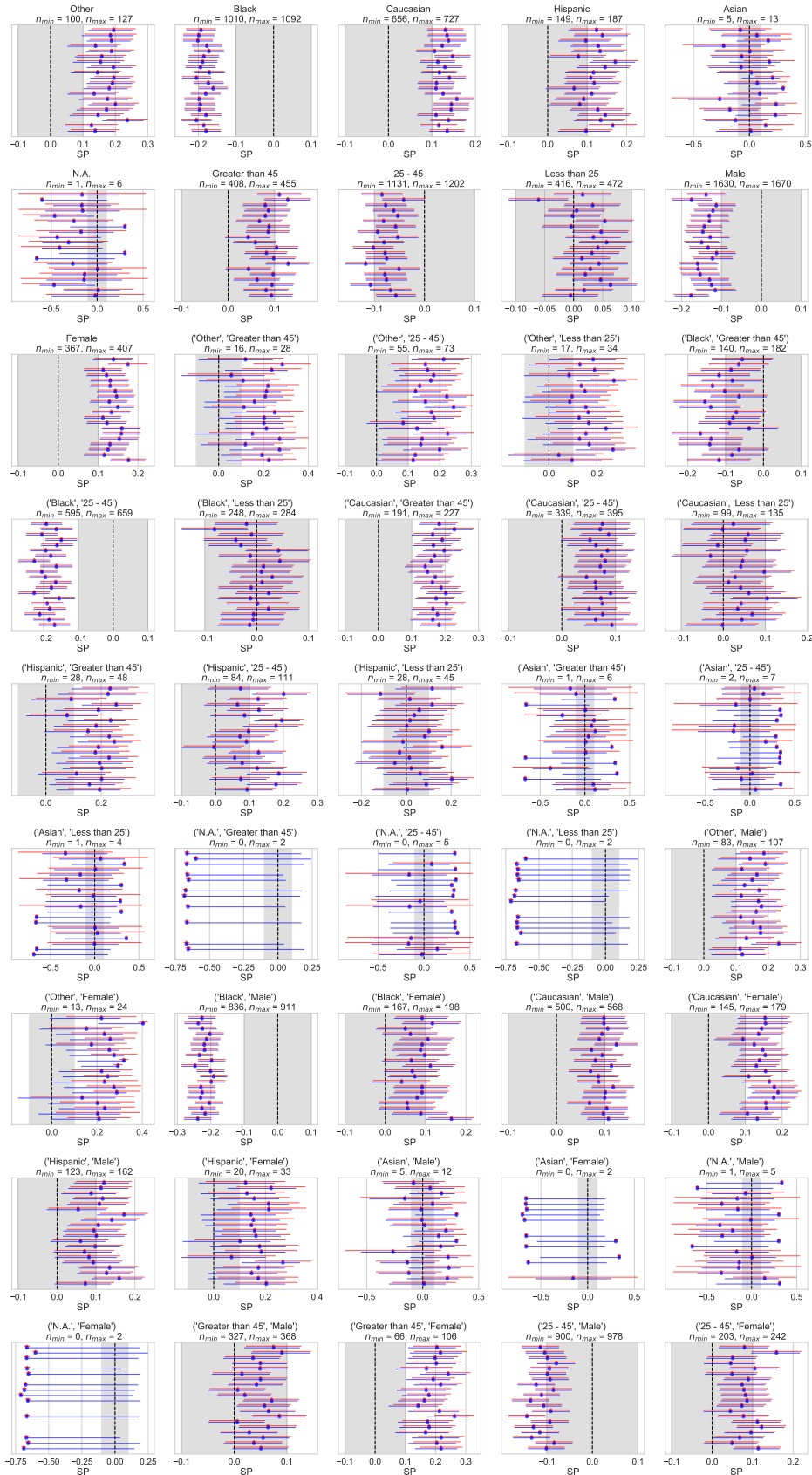

Figure 8: Point-wise estimation versus confidence intervals, COMPAS dataset.

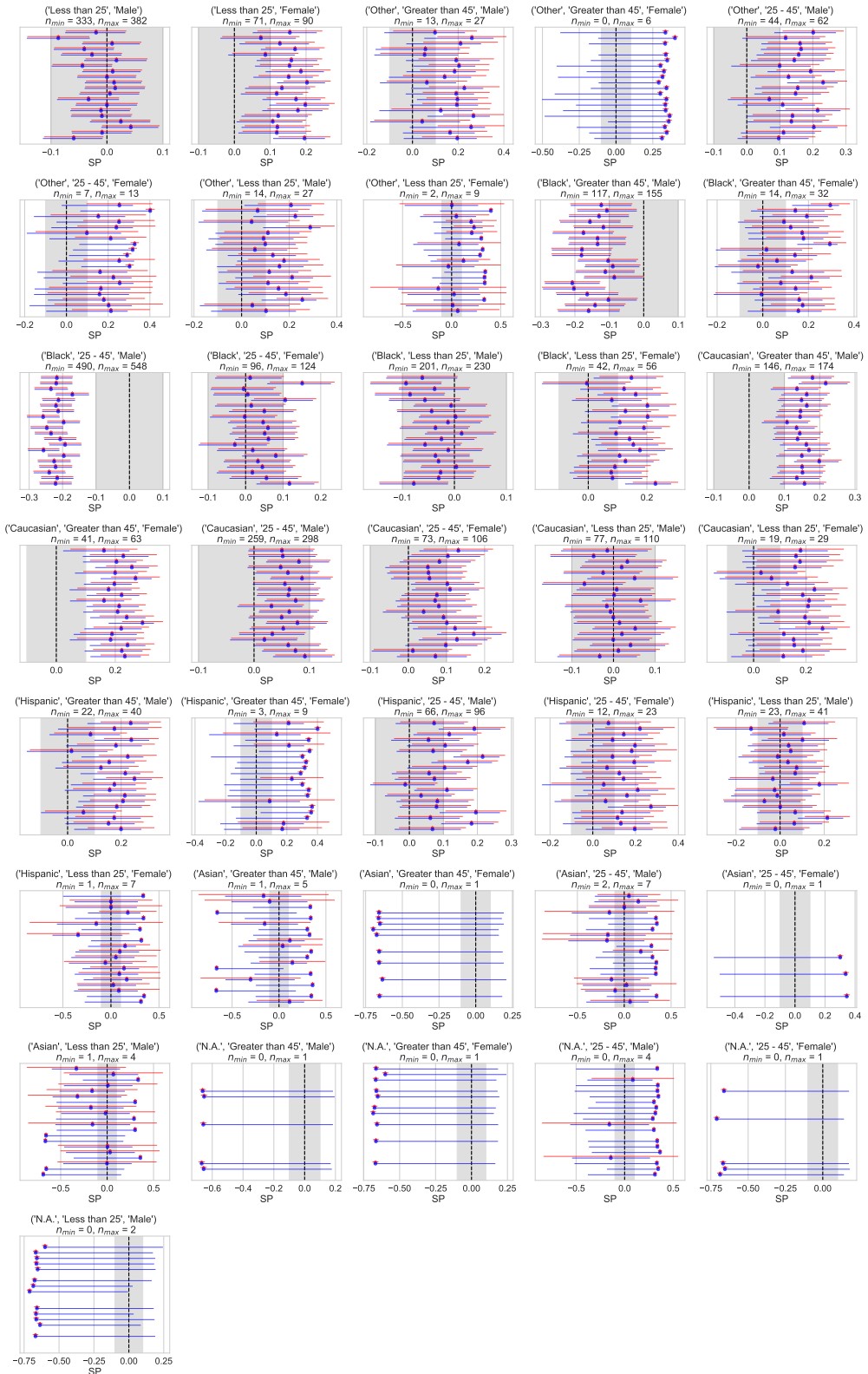

Figure 9: Point-wise estimation versus confidence intervals, COMPAS dataset (continued).

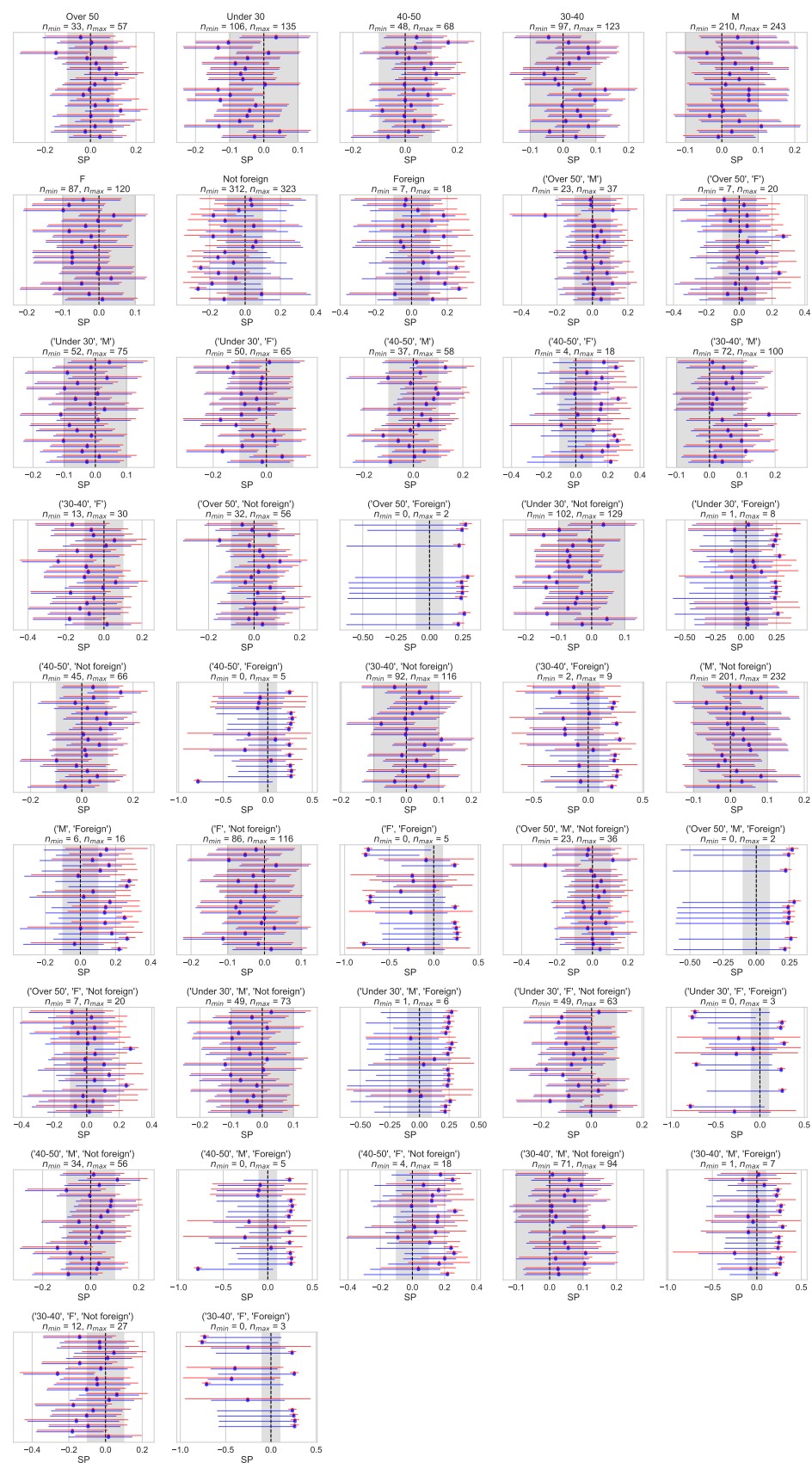

Figure 10: Point-wise estimation versus confidence intervals, German dataset

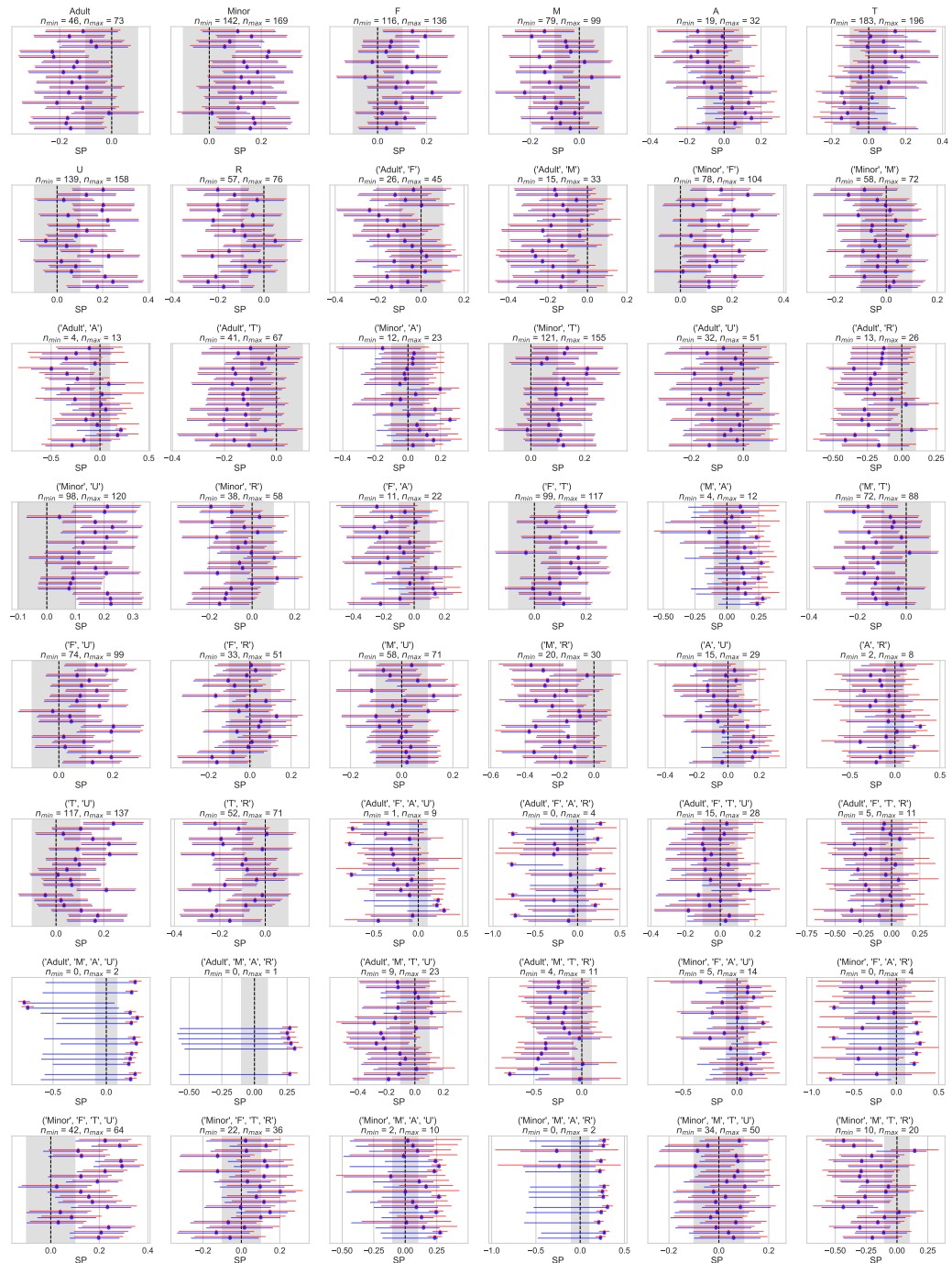

Figure 11: Point-wise estimation versus confidence intervals, Student dataset

Furthermore, in Figure 4, we considered all intersectional groups (across all 20 train-test splits) on COMPAS and Adult to compare the decision from our framework with those from a threshold-based $\gamma$SP approach. In Figure 12, we extend the same approach for the two other datasets, German and Student. The results show that a fixed threshold on $\gamma$SP fails to consistently distinguish fairness violations, highlighting the necessity of a size-adaptive hypothesis-testing approach like SAFT.

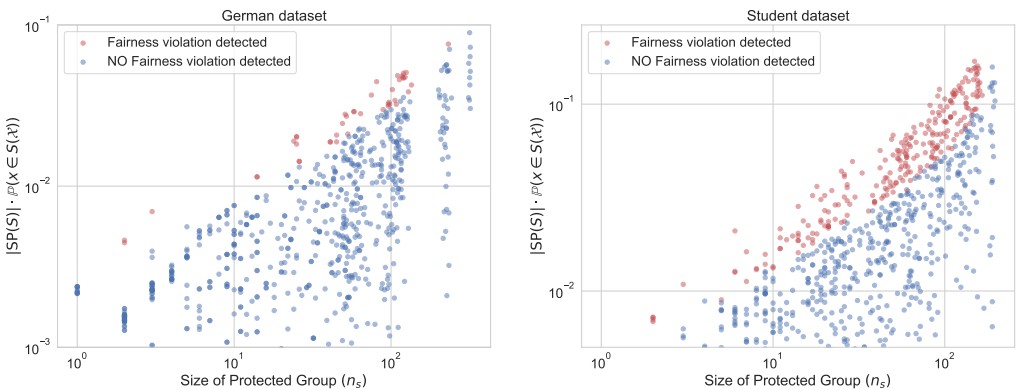

Figure 12: Protected groups size, $\gamma$SP scores, and interval-based fairness violations.

## C.4 Bayesian vs. Asymptotic Interval Convergence

To illustrate how our Bayesian credible intervals smoothly interpolate between robust small-sample uncertainty and asymptotic normality, Figure 13 shows four synthetic subgroups with sizes $n \in \{1, 10, 100, 1025\}$. In each panel we plot (i) the posterior density of the statistical-parity gap under a uniform Dirichlet prior (blue curve), (ii) the corresponding 95% credible interval (blue shaded region), and (iii) the asymptotic 95% confidence interval from Theorem 1 (red curve region) centered at the same point estimate. As $n$ grows, the two intervals converge and the posterior density sharpens, while for very small $n$ the Bayesian interval remains appropriately wide, reflecting high epistemic uncertainty.

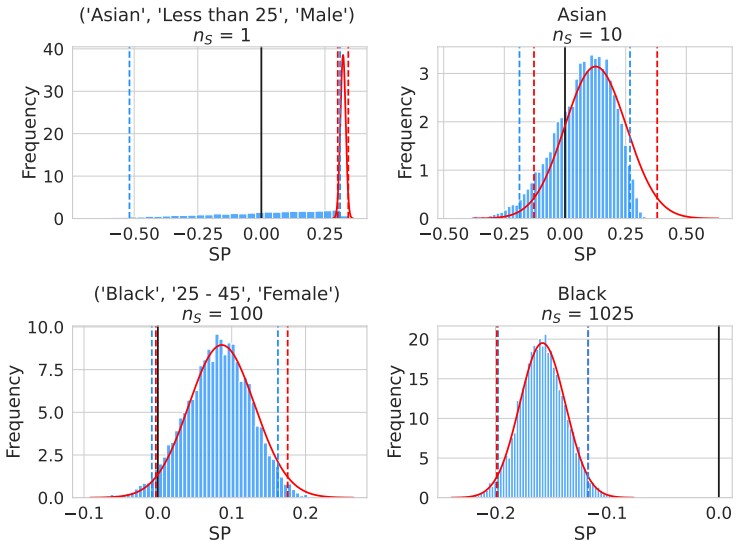

Figure 13: Comparison of Bayesian credible intervals (blue) and asymptotic normal confidence intervals (red) for subgroups of size $n_S$. As $n_S$ increases, the posterior density concentrates and the two intervals converge; for small $n_S$, the Bayesian interval remains larger, correctly encoding high uncertainty.

## C.5 Effects of small sample size on the asymptotic results

The Wald variance for subgroup $\mathcal{S}$ can be rewritten as:

$$\sigma^2 = \frac{\frac{p_{1,S}}{p_{1,S}+p_{0,S}}\left(1 - \frac{p_{1,S}}{p_{1,S}+p_{0,S}}\right)}{n_S} + \frac{\frac{p_{1,\bar{S}}}{p_{1,\bar{S}}+p_{0,\bar{S}}}\left(1 - \frac{p_{1,\bar{S}}}{p_{1,\bar{S}}+p_{0,\bar{S}}}\right)}{n_{\bar{S}}}$$

Since $n_S \approx n \cdot P(S)$ (and analogously for $n_{\bar{S}}$) the variance decreases at rate $1/n$, even for rare groups. Both asymptotically and for small samples, small subgroups do not lead to inflated variance.

That said, the more subtle issue is that in finite samples, the plug-in variance $\sigma^2$ may actually underestimate uncertainty when $n_S$ is small. In particular, when the estimate of $\frac{p_{1,S}}{p_{1,S}+p_{0,S}}$ is very close to 0 or 1, the estimation of the term $\frac{p_{1,S}}{p_{1,S}+p_{0,S}}\left(1 - \frac{p_{1,S}}{p_{1,S}+p_{0,S}}\right)$ in the variance formula becomes near zero. This drives the Wald variance estimate to be artificially small, even though the underlying sampling uncertainty is high for such extreme proportions with small $n_S$.

This can make the Wald test *anti-conservative*, falsely rejecting the null due to high variance in small groups not captured by the asymptotic approximation. This effect can be seen, for example, in the top-left plot of Figure 13.

This is why we introduce the Bayesian Dirichlet–multinomial test to provides calibrated inference in the small-sample regime. Hence, the problem is not variance inflation, but undercoverage due to inappropriate reliance on asymptotics when subgroup counts are small.

## C.6 Choice of $\tilde{n}$

In Algorithm 1 we suggested choosing $\tilde{n} = 30$, since thirty observations is a widely used rule-of-thumb to trust CLT-based Wald tests [38] because coverage and power stabilize around that cell size in typical multinomial settings. In Table 2 we analyze the effect of the choice of $\tilde{n}$ for SAFT applied to the Adult dataset. One can observe how from around twenty observations the total number (over all train-test splits and over all possible sub-groups) of fairness violations detected stabilizes. Hence, also our empirical results confirm thirty observations as a safe choice to switch from the bayesian to the asymptotic test. With a smaller $\tilde{n}$, e.g., $\tilde{n} = 5$, the amount of samples appears to be too small to rely on asymptotic results, indeed the CLT might induce to over-reject the null, as explained in Appendix C.5.

Table 2: Total number of fairness violations detected by SAFT in Adult for all train-test splits and for all sub-groups for different values of $\tilde{n}$

| $\tilde{n}$ | 1 | 2 | 5 | 10 | 20 | 30 | 40 | 50 | 100 | 200 | 500 | 1000 |
|---|---|---|---|---|---|---|---|---|---|---|---|---|
| # of violations | 1159 | 1144 | 1118 | 1098 | 1097 | 1097 | 1097 | 1096 | 1098 | 1097 | 1097 | 1097 |

Clearly, users may raise or lower $\tilde{n}$ if their data warrant it.

## C.7 Influence of the prior

We now explore the impact of the prior choice and the incorporation of prior information, derived from the training set. We consider:

- Uniform (flat) Dirichlet prior, which imposes no substantive prior information;
- Empirical Bayes priors derived from training-set frequencies.

For each subgroup we set

$$\alpha = \frac{\lambda}{\max(1, \min(c_{0,s}, c_{0,\bar{s}}, c_{1,s}, c_{1,\bar{s}}))}[c_{0,s} + 1, c_{0,\bar{s}} + 1, c_{1,s} + 1, c_{1,\bar{s}} + 1], \quad \lambda \in \{0.5, 1, 2, 5\}$$

where $c_{i,j}$ is the number of training samples with label $i \in \{0, 1\}$ in group $j \in \{s, \bar{s}\}$.

This construction scales the prior strength by $\lambda$ while preserving the empirical class proportions, and caps the influence when any cell count is extremely small. Table 3 shows the results with the confidence/credible intervals (CIs) for some selected subgroups from the Adult dataset.

The results show prior influence for extremely small subgroups, while all CIs rapidly converge to the asymptotic regime as $n_S$ grows. For the smaller $n_S$ case, the Wald intervals are spuriously narrow, clearly under-representing the true sampling uncertainty. In contrast, the uniform-prior Bayesian credible intervals are much larger, appropriately reflecting the scarcity of data. As we introduce empirical-Bayes priors, these intervals shrink toward the Wald width but remain more conservative than asymptotic bounds, demonstrating that modest data-informed priors can stabilize inference without overconfidence. With hundreds or thousands of samples, all methods produce nearly identical intervals, illustrating that prior effects vanish once more data is available.

Table 3: Influence of the prior

|  | Other, Under 18, Female | A.P.I., Over 65, Male | White, Over 65, Female | Black |
|---|---|---|---|---|
| $n_S$ | 2 | 12 | 210 | 1522 |
| Asymptotic normal CI | [-0.204, -0.192] | [-0.271, 0.041] | [-0.182, -0.123] | [-0.130, -0.097] |
| Bayesian CI with uniform prior | [-0.190, 0.488] | [-0.176, 0.176] | [-0.174, -0.115] | [-0.127, -0.097] |
| Bayesian CI with empirical prior, $\lambda = 0.5$ | [-0.240, 0.014] | [-0.196, 0.048] | [-0.179, -0.121] | [-0.138, -0.114] |
| Bayesian CI with empirical prior, $\lambda = 1$ | [-0.236, 0.037] | [-0.196, 0.052] | [-0.179, -0.123] | [-0.138, -0.113] |
| Bayesian CI with empirical prior, $\lambda = 2$ | [-0.228, 0.035] | [-0.177, 0.037] | [-0.177, -0.125] | [-0.138, -0.114] |
| Bayesian CI with empirical prior, $\lambda = 5$ | [-0.195, 0.000] | [-0.156, 0.011] | [-0.176, -0.128] | [-0.139, -0.113] |
| Bootstrapping CI | [-0.204, -0.192] | [-0.203, 0.073] | [-0.178, -0.118] | [-0.130, -0.098] |

## C.8 Comparison with Bootstrapping

In moderate to large subgroups, nonparametric bootstrap intervals (e.g., [44]) coincide, both theoretically and empirically, with the Wald limits obtained from our CLT analysis, so bootstrapping adds computation without accuracy gains at that scale. In small intersectional subgroups where Wald approximations fail, the bootstrap typically inherits the same pathologies: with pronounced class imbalance, resamples frequently yield zero counts in one or more contingency cells (e.g., no predicted positives for group $S$), producing degenerate or undefined disparity estimates and a highly discrete, skewed resampling distribution that yields off-center, overly narrow intervals. In the last row of Table 3, the bootstrap intervals closely track the asymptotic confidence interval even at small $n_S$, indicating that resampling does not remedy the underlying approximation error in this setting. Accordingly, we use analytic CLT intervals when regularity conditions are met, and Dirichlet-multinomial credible intervals when any cell counts are in the single digits, where they remain well-calibrated and avoid degeneracy.

## C.9 Results for Equal Opportunity

In Figures 14 and 15 (Adult dataset) and Figures 16 and 17 (COMPAS dataset), we present the results for Equal Opportunity. In each panel, $n_{min}$ and $n_{max}$ denote the minimum and maximum number of instances belonging to the protected group with $y = 1$ across different train–test splits. Compared to the plots for Statistical Parity Figures 6 to 9, those for Equal Opportunity show wider CIs, consistently reflecting lower $n_{i,j}$ counts given the additional conditioning on $y = 1$ for Equal Opportunity.

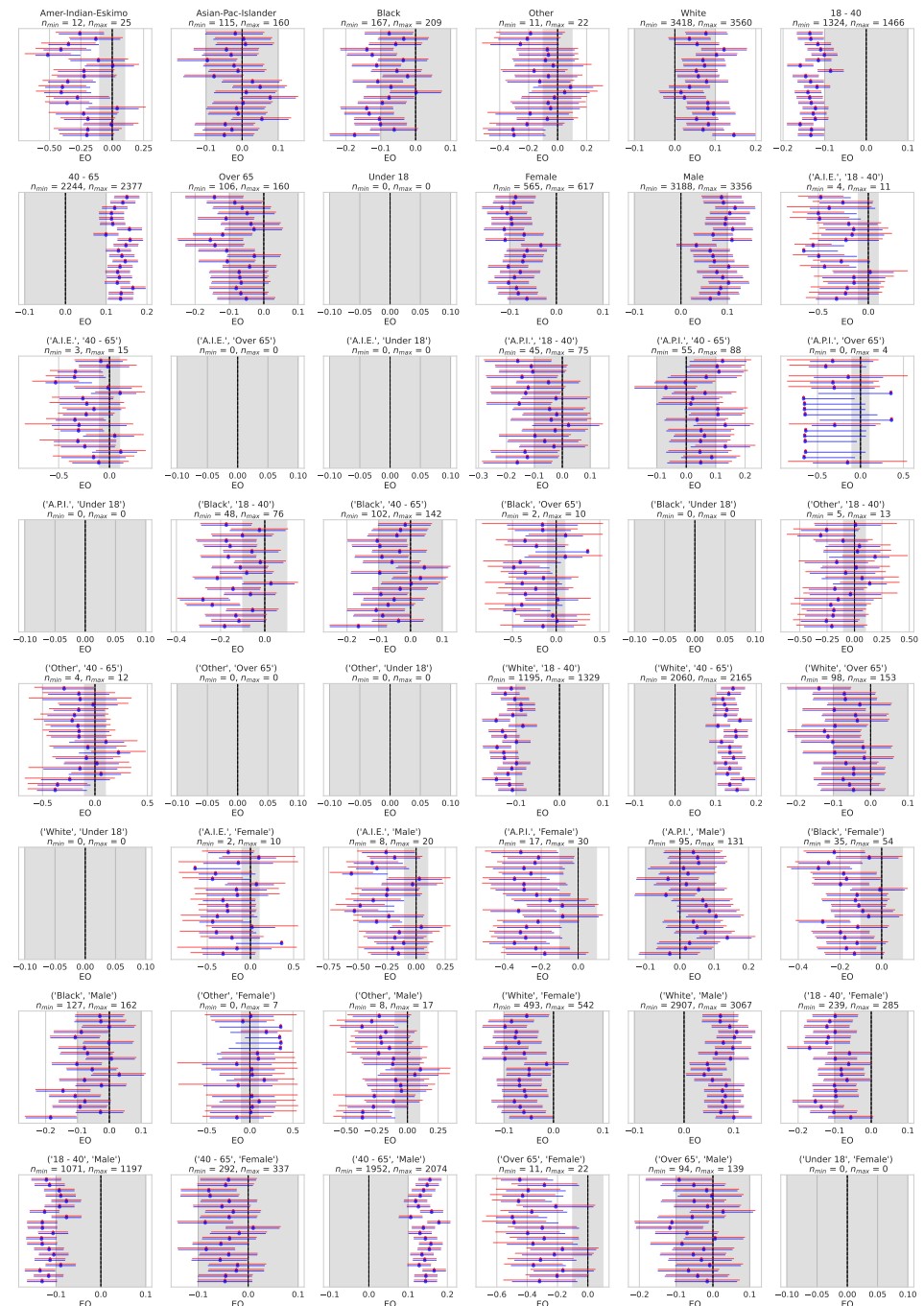

Figure 14: Point-wise estimation versus confidence intervals for Equal Opportunity, Adult dataset.

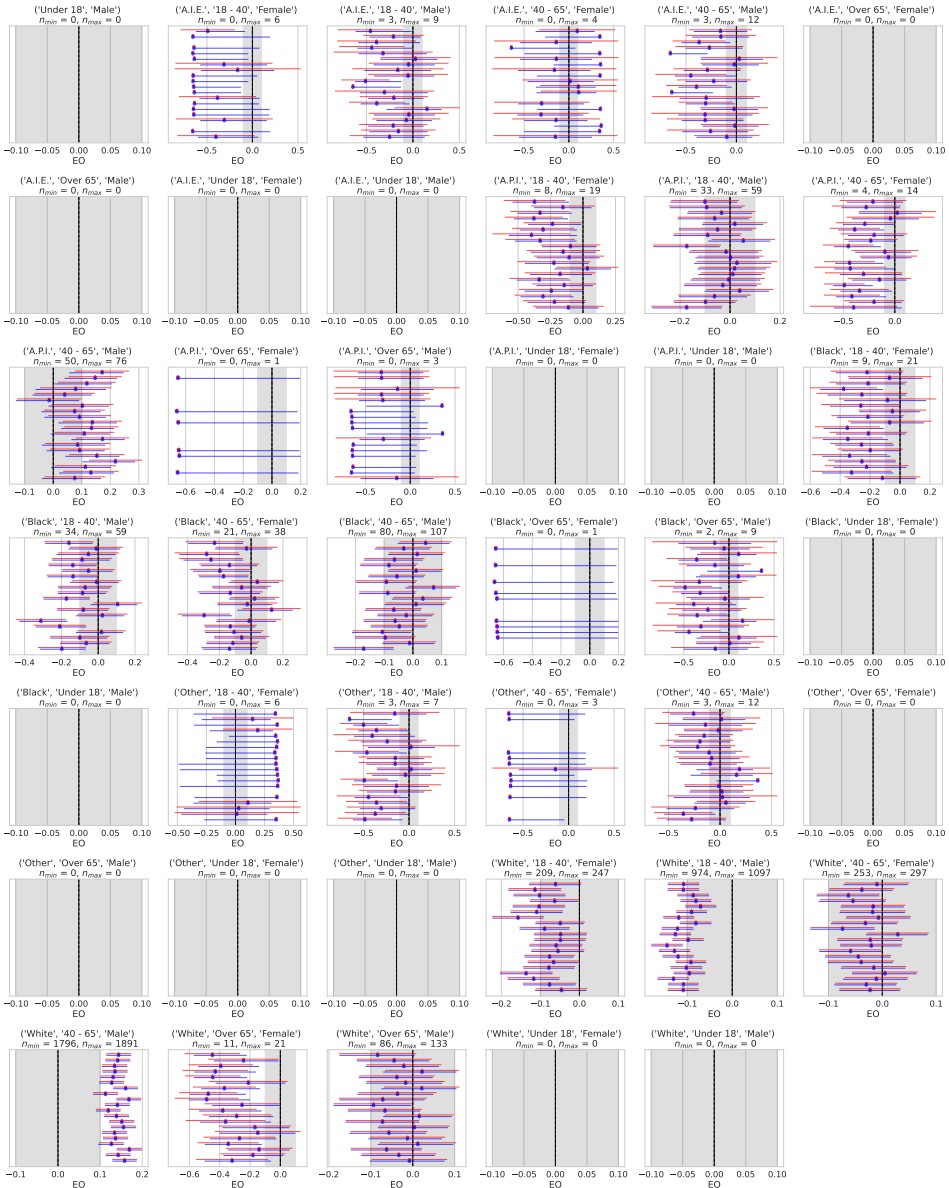

Figure 15: Point-wise estimation versus confidence intervals for Equal Opportunity, Adult dataset (continued).

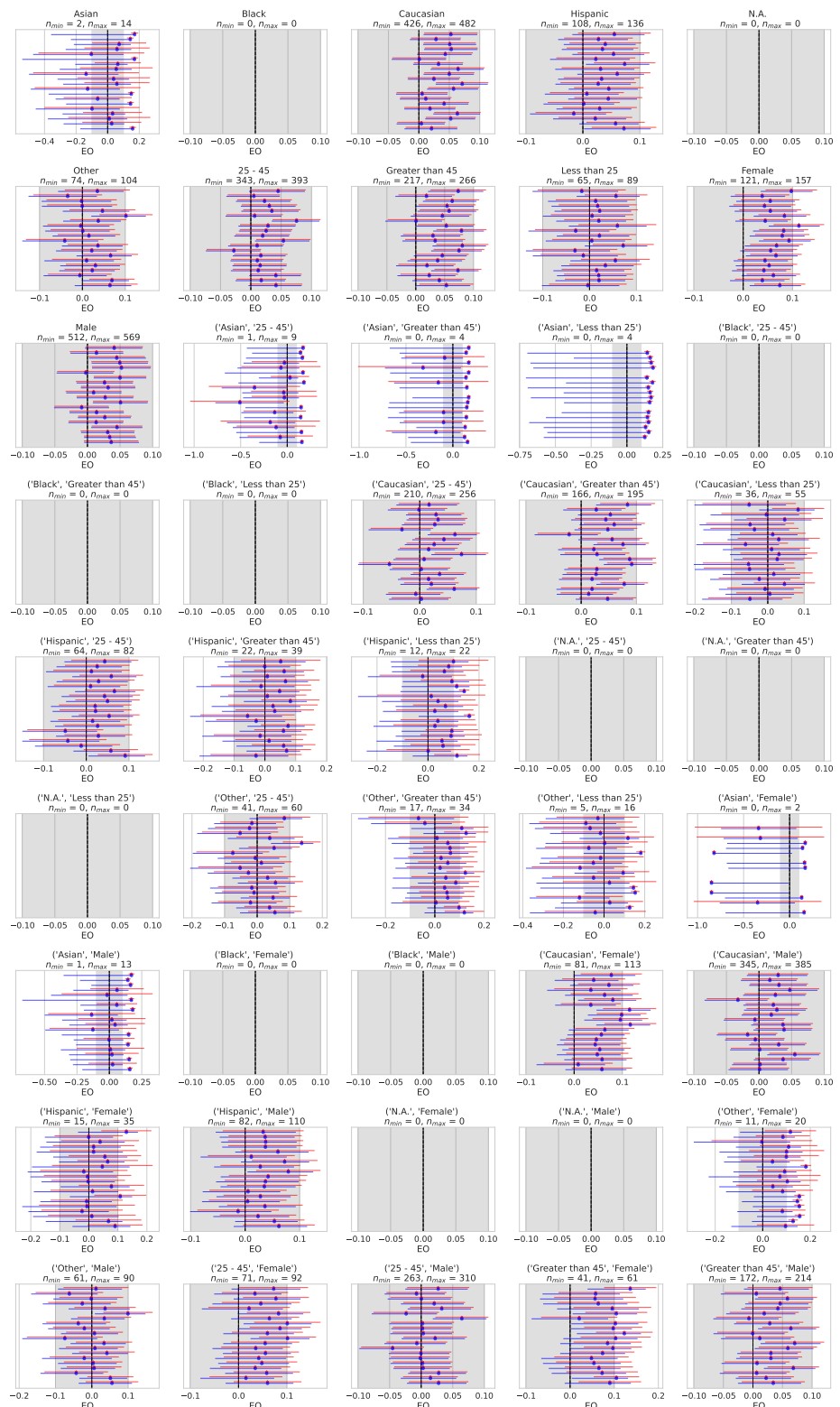

Figure 16: Point-wise estimation versus confidence intervals for Equal Opportunity, COMPAS dataset

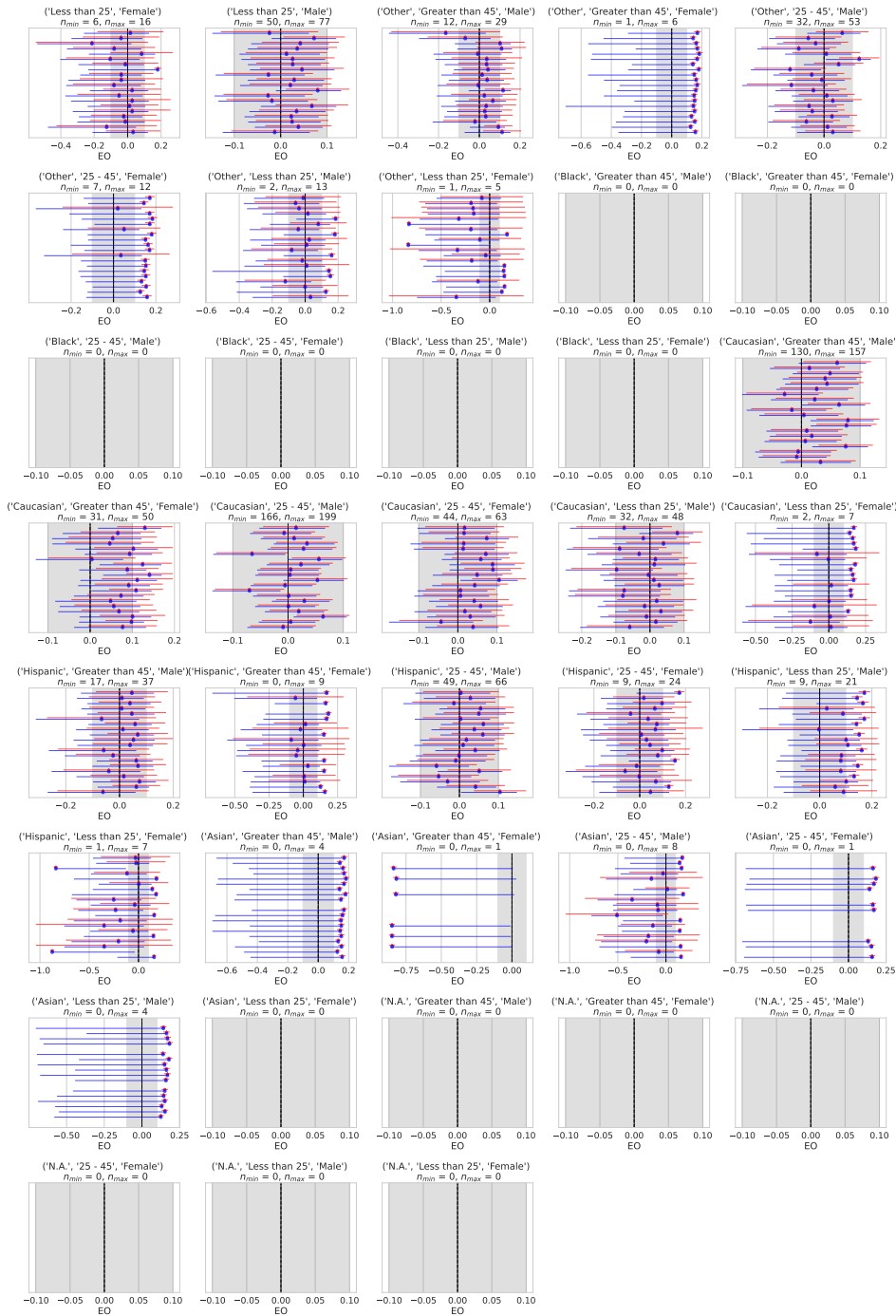

Figure 17: Point-wise estimation versus confidence intervals for Equal Opportunity, COMPAS dataset (continued)

