# OpenReview forum: "Size-adaptive Hypothesis Testing for Fairness"
_NeurIPS.cc/2025/Conference — NeurIPS 2025 poster_

### Official Review · Reviewer_ujF4 · 2025-06-16

**Clarity:** 4
**Significance:** 2
**Originality:** 1
**Rating:** 3
**Confidence:** 5

**Summary:**

The authors propose an adaptative way for testing whether a fairness criterion holds. Instead of using a pre-defined threshold, a formal hypothesis test is run, which adapts naturally to the number of samples available in different protected groups. The main focus of the paper is on demographic (statistical) parity, although some extensions are possible by applications of the Delta-method. Specifically, for relatively large samples an asymptotically valid Wald test is proposed, while for small samples a Bayesian approach with a conjugate Dirichlet prior is used.

**Questions:**

- (Q1) is there a need for incorporating uncertainty in the constructed classifier f? In the current treatment, f seems to be treated as if it is not derived based on data?
- (Q2) Is there a way of performing a similar test using bootstrap, in a non-parametric fashion?

**Ethical Concerns:**

["NO or VERY MINOR ethics concerns only"]

**Final Justification:**

The proposed method, which performs hypothesis testing for statistical parity, is really not a novel idea. Many works in the literature perform such hypothesis tests when doing experiments; these works, however, do not claim the hypothesis-testing to be a novel contribution.

Furthermore, if a novel way of testing such fairness hypotheses was proposed, that would be great. However, this is not the case either -- the results are a direct corollary of classical results in statistics.

Therefore, in my view, this paper should not be accepted. Having said this, the paper is well-written, and may be a useful educational note for some, so I am not very strongly against its acceptance (hence the score 3)

**Limitations:**

Yes.

**Paper Formatting Concerns:**

No formatting concerns.

**Quality:**

3

**Strengths And Weaknesses:**

Strengths:

- (S1) The notation is clear, nice, and easy to follow,
- (S2) The paper is well-written, and the concepts are presented clearly,
- (S3) A formal hypothesis test approach is adopted, whereas common approaches may use a pre-defined threshold.

Weaknesses:

While acknowledging the positives listed above, the major drawback is that all of the theory appearing in the paper is really well-known and widely used in the statistics literature, so there is no need for new theoretical developments (Theorem 1 follows from a variant of the CLT, Theorem 2 by the Delta method). In addition, the adaptation of the testing to the context of fairness does not require a huge amount of translation. Therefore, this paper presents a nice technical note, but does not reach the level expected from a NeurIPS contribution. Since the basic setting of demographic parity is mastered, one obvious way to improve is to study a more ambitious problem, such as conditional or causal variants of parity, which would likely provide more significant theoretical and methodological challenges.

---

> ### Author Rebuttal · Authors · 2025-07-30
>
> ### **Originality and Significance for NeurIPS**
>
> We thank the Reviewer for clearly acknowledging the clarity and the technical quality of our work. On the other hand, Reviewer ujf4 believes that **Significance** and **Originality** of our work are low and do not reach the bar for NeurIPS. We respectfully disagree with this assessment and in the following we discuss why our contribution does not amount only to repackaging standard statistics.  Classical tools such as the multivariate CLT and the Delta method are indeed decades old; however, their *application, calibration, and validation* for high-resolution fairness auditing is not.
>
> To the best of our knowledge, no existing work provides a standardized, statistically calibrated test for fairness violations that adapts to subgroup size and gives both asymptotic and finite-sample guarantees. Current methods (δ-SP, γ-SP, etc.) rely on fixed thresholds with no error control. Our proposal fills this methodological gap and is designed to become a standard testing protocol for fairness auditing, analogous to how χ² or t‑tests are used in classical statistics.
> We would like to point out that both Theorem 1 and 2 follow from a smart use of combined CLT, Delta Method and Slutsky theorem, and the simple formulation follow from a refined choice of combining probabilities of disjoint events from which we observe that many practically used fairness metrics are derivable and which allows for an easy computation of the covariance matrix.
>
>
> Moreover, the Reviewer writes:
> >*‘’Since the basic setting of demographic parity is mastered, one obvious way to improve is to study a more ambitious problem, such as conditional or causal variants of parity, which would likely provide more significant theoretical and methodological challenges.’’*
>
> Statistical parity is not “mastered” in real deployments; regulators and practitioners still rely on threshold choices and point estimates.  We chose it as a canonical case to establish the statistical foundations.  However, our framework is *metric- and model- agnostic* and readily extends to other fairness metrics.  Conditional or causal notions are indeed exciting directions, and our resolution-limit analysis provides the very diagnostic tool needed before estimating causal effects in sparse conditions. Specifically,  Theorem 2 already generalizes to the mentioned conditional variants.
>
> NeurIPS values contributions that close the gap between statistical theory and machine learning practice in societally critical settings.  By unifying analytic variance, Bayesian calibration, and a decision-theoretic resolution limit, then demonstrating their impact on real datasets, we deliver a rigorously grounded, practically deployable method that the fairness-in-ML community still lacks. In these regards, the scores given by the Reviewer to Originality and Significance, seem not to be sustained.
>
> Finally, the Reviewer gave a general rating of *2: Reject [For instance, a paper with technical flaws, weak evaluation, inadequate reproducibility and incompletely addressed ethical considerations]*, which does not appear to be adequately sustained by the arguments in the review. Our work offers a rigorously grounded, ready-to-use tool that addresses an urgent need in responsible AI practice, and thus meets the bar for significance and relevance at NeurIPS.
>
> ### **Regarding uncertainty in the constructed classifier $f$**
>
> We thank the reviewer for raising this important point. In our framework, the classifier $f$ is indeed derived from data, but it is treated as a fixed, given object at the moment of fairness auditing. This assumption reflects a common operational scenario: fairness audits are typically conducted *after* model training, on a deployed or pre-trained classifier whose predictions are to be evaluated for fairness.
>
> Our methodology focuses on quantifying the **statistical uncertainty in fairness metrics**, arising from the finite sample size of the evaluation data, especially in small or rare subgroups. The uncertainty we account for is thus epistemic, stemming from sampling variability rather than model uncertainty.
>
> That said, we agree that in settings where the classifier outputs **probabilistic predictions**, it would be possible and potentially useful to incorporate model uncertainty into the fairness analysis. For instance, one could assess fairness over distributions of predicted probabilities, or incorporate uncertainty from stochastic or Bayesian classifiers into the fairness tests. This is a promising direction, and one that complements our work: our current methodology can serve as a principled post hoc test of fairness, while future extensions could integrate upstream uncertainty in the predictions of $f$ itself.
>
> In summary, our approach does not require the classifier to express uncertainty, but is compatible with extensions that incorporate such uncertainty when available. We will integrate this discussion in the manuscript and we plan to explore it in future work.
>
>
> ### **Bootstrap intervals: why they fall short in sparse intersectional settings**
> Thank you for raising this important point. We experimented extensively with non-paramentric bootstrap intervals prior to the submission and decided not to include them because, in our settings, bootstrap often fails to offer any advantages over either the frequentist or the Bayesian approaches. However, we agree that it would be better to include a discussion about this point in the paper anyway.
>
> We highlight now some characteristics of the bootstrap approach. First, in moderate to large subgroups, bootstrap intervals converge both theoretically and empirically to the same Wald limits delivered by our analytic CLT. At that scale, bootstrap therefore adds only computational overhead without any improvement. In the small $n_s$ intersectional slices where analytic Wald intervals break down, bootstrap inherits the same weaknesses. Resampling with replacement in class unbalanced scenarios frequently generates samples in which one of the four contingency cells is zero (e.g., no predicted positives for group $S$), producing degenerate or undefined estimates of $SP$. Moreover, the resulting empirical distribution is highly discrete and strongly skewed, which yields uncentered confidence intervals. For these reasons, we opted for the analytic CLT, whose validity can be proved, and for the Dirichlet–multinomial credible interval, which remains calibrated even when cell counts are in the single digits. Nonetheless, the camera‑ready version will include a dedicated appendix comparing our method with the bootstrap approach, documenting the pathologies above and clarifying the regimes in which bootstrap is a viable alternative.
>
> ---
>
> We hope these clarifications address the Reviewer’s concerns.
> We would be grateful to engage the Reviewer in a fruitful discussion and to provide further details if needed.

---

> > ### Comment · Reviewer_ujF4 · 2025-08-04
> >
> > I thank the authors for their detailed response. Here are some points I wish to add, to conclude the discussion:
> >
> > - I have checked the proofs carefully, and I am familiar with the mentioned literature. I maintain that the derived results do not require sophisticated theoretical developments, even though the authors may see this differently.
> >
> > ---
> >
> > > Statistical parity is not “mastered” in real deployments; regulators and practitioners still rely on threshold choices and point estimates.
> >
> > Here, I guess the word mastered may be ambiguous. What is certainly the case is that a number of papers in the literature already perform hypothesis testing for verifying if statistical parity holds (although the most common approach may be based on bootstrap). Claiming that the literature is entirely unaware of doing hypothesis tests seems like a rather strong statement.
> >
> > Therefore, I maintain that the novelty in the paper is limited.
> >
> > ---
> >
> > > However, our framework is metric- and model- agnostic and readily extends to other fairness metrics.
> >
> > This statement is not supported by the current work (where very few metrics are considered). Generally, it is useful to provide arguments which are supported.
> >
> > ---
> >
> > Finally, as already said, I view this as a useful note. Having read the authors feedback, I acknowledge that a score of 3 is more appropriate than 2, so I have adjusted it accordingly. I certainly see the reasons to reject as stronger (lack of novelty) than to accept (good clarity, educational content for some part of the audience).

---

> > > ### Author Response · Authors · 2025-08-05
> > > **Response to Reviewer Follow-Up**
> > >
> > > We thank Reviewer ujf4 for the thoughtful discussion and for rising their score to a 3, reflecting their assessment of the paper's technical soundness and clarity. We sincerely appreciate the serious and deep discussion, as well as the professionality and engagement shown by the Reviewer in this process.
> > >
> > > We recognize that the remaining disagreement is not about the correctness of our work, but rather a good-faith difference in perspective on what constitutes a novel contribution for NeurIPS. In this regard, we refer to the **NeurIPS 2025 Reviewer Guidelines:**
> > >
> > > > *Originality:* Does the work provide new insights, deepen understanding, or highlight important properties of existing methods? Is it clear how this work differs from previous contributions, with relevant citations provided? Does the work introduce novel tasks or methods that advance the field? Does this work offer a novel combination of existing techniques, and is the reasoning behind this combination well-articulated? **As the questions above indicates, originality does not necessarily require introducing an entirely new method. Rather, a work that provides novel insights by evaluating existing methods, or demonstrates improved efficiency, fairness, etc. is also equally valuable.**
> > >
> > >
> > > ### About Originality
> > >
> > > Reviewer ujf4 correctly notes that hypothesis testing for fairness has been explored in the literature, often using bootstrap methods. Our paper’s primary contribution is not the general idea of hypothesis testing, but the development of the **first standardized, statistically calibrated protocol specifically designed for high-resolution and intersectional fairness auditing, which provides guarantees precisely where existing methods fail.**
> > >
> > > As detailed in our paper and rebuttal, standard bootstrap methods become unreliable or degenerate in the sparse-data regimes common to intersectional fairness. Our work addresses this critical methodological gap by providing:
> > > 1. an analytic CLT-based test for larger subgroups,
> > > 2. a calibrated Bayesian approach for sparse subgroups, and
> > > 3. a decision-theoretic 'resolution limit' to guide practitioners. This unified framework is the core novelty and is absent from prior work.
> > >
> > > ---
> > >
> > > ### About Unsupported Generality
> > >
> > > We would like to clarify that our asymptotic result (Theorem 2) and the Bayesian framework are designed to **generalize by construction** to any fairness criterion that can be expressed as a smooth function of the confusion matrix and/or related conditional probabilities. Indeed, our framework can model several common definitions of group fairness, including the first 13 definitions of table 1 in *'Fairness definition explained'* [Verma and Rubin, 2018]. We will clarifiy this in the paper and we will include a dedicated appendix where we explicitly walk through the application of our framework to another key metric, such as Equalized Odds, demonstrating how the underlying machinery directly extends.
> > >
> > > ---
> > >
> > > ### Conclusions
> > >
> > > While Reviewer ujf4 frames our work as 'educational,' we believe this perspective overlooks the significant value of establishing foundational, rigorous statistical tools for a field as critical as AI Fairness. The ML community still lacks reliable, off-the-shelf protocols for fairness auditing that practitioners and regulators can trust. Our work delivers exactly such a protocol. We contend that building these crucial bridges between statistical theory and ML practice is a core mission of NeurIPS.
> > >
> > > In summary, we believe this paper is technically sound, clearly presented, and addresses a well-documented and urgent need in the responsible AI community. The central disagreement with Reviewer ujf4 boils down to a narrow definition of novelty that prioritizes sophisticated new theory over the significant contribution of creating a rigorous, unified, and practical methodology that solves a known problem where prior art falls short.

---

### Official Review · Reviewer_L4Jh · 2025-07-02

**Clarity:** 3
**Significance:** 2
**Originality:** 2
**Rating:** 4
**Confidence:** 4

**Summary:**

The paper introduces a new framework to measure fairness based on hypothesis testing instead of comparing a point estimate of the fairness metric with a predefined threshold. They provide two tests to detect statistical parity unfairness. These tests consider the uncertainty and group sizes due to finite test sets. The first test utilizes the confidence intervals based on asymptotic normal approximation of disparity; this test is valid for large sample sizes and based on CLT. When the sample sizes are small for subgroups, the authors propose another test that utilizes Bayesian approach. They propose a Dirichlet prior on the estimated values for the statistical parity, then get samples from the posterior and build an empirical confidence interval. The authors also demonstrate how the methodology can be generalized for other fairness notions, such as equal opportunity.  The numerical experiments cover 4 real datasets, and the proposed method is compared against the fixed threshold rule and the scaled(by the probability of the subgroup) threshold rule.

**Questions:**

Please see the weaknesses section.

**Ethical Concerns:**

["NO or VERY MINOR ethics concerns only"]

**Final Justification:**

I think that the authors are providing detailed answers to the questions raised by the reviewers. They provided new results regarding the sensitivity to prior selection for the small sample tests, and they provided new results that provide comparisons with an existing baseline method bootstrapping that provides confidence intervals for fairness violations.

I still think that the technical contribution/novelty of the paper could be considered limited, but overall, the positive sides of the paper slightly overweight the limitations.

**Limitations:**

Yes, authors addressed the limitations and broader impact.

**Paper Formatting Concerns:**

No formatting issues.

**Quality:**

3

**Strengths And Weaknesses:**

Strengths

- The idea of using hypothesis testing for unfairness detection is novel; the framework unifies both frequentist and Bayesian approaches and switches between them based on the test data availability.
- The authors provide empirical results on four real-world datasets. The proposed method avoids false alarms and detects unfairness that threshold-based rules miss.
- The paper is well-structured and easy to follow. The mathematical derivations are complete.

Weaknesses

- The technical contributions of the paper are limited; the contribution of the paper is mostly proposing hypothesis testing for fairness detection.
- It is not clear how the small subgroup sizes affect the first proposed test. The variance estimate would be inflated due to small subgroups, probably, but a clear explanation of this could be beneficial before introducing the second test.
- For the Bayesian approach, the authors do not report on the sensitivity of the prior selection. It could be interesting to see priors informed by the training set frequencies.
- The experiments section can have better baselines, such as other confidence interval generation methods, i.e., bootstrap intervals available in FairLearn. The extension for the other fairness notions is explained in the paper, but the experiments section only focuses on statistical parity.

---

> ### Author Rebuttal · Authors · 2025-07-31
>
> Thank you for the thoughtful and constructive feedback. Below we address each weakness in detail and describe the revisions we will incorporate to strengthen the paper.
>
> ---
>
> **W1:** To the best of our knowledge, in the fairness literature, there is no universally accepted analogue of the χ², t-, or log-rank tests that offers to practitioners a standardized, statistically calibrated test for detecting fairness violations that adjusts to subgroup size while providing both asymptotic and finite-sample guarantees. What exists instead are score heuristics such as δ-SP or γ-SP whose numerical thresholds (±0.05, ±0.10, etc.) are chosen ad-hoc and carry no type-I/type-II guarantees. Regulators, standards bodies, and applied ML teams have repeatedly called for **principled, statistically grounded decision rules** that can withstand legal and scientific scrutiny.
>
> Our method addresses this gap and is intended to serve as a standard protocol for fairness auditing, in the same way that χ² or t‑tests function in classical statistics. We note that Theorems 1 and 2 result from an effective combination of the Central Limit Theorem, the Delta Method, and Slutsky’s theorem, while the concise formulation arises from a careful way of combining probabilities of disjoint events. This not only shows that many commonly used fairness metrics can be derived within our framework but also enables straightforward computation of the covariance matrix.
>
> ---
>
> **W2:** We thank the Reviewer for pointing out the need for more clarity on how small subgroup sizes affect our first test (the Wald test).
>
> The concern raised is that small subgroup sizes might inflate the variance estimate, leading to conservative behavior. However, under the asymptotic regime in Theorem 1, this is not the case. The Wald variance estimate for subgroup $S$ can be rewritten as:
>
> $\hat{\sigma}^2 = \frac{p̂_S (1 - p̂_S)}{n_S}$
>
> $+ \frac{p̂_\bar{S} (1 - p̂_\bar{S})}{n_{\bar{S}}}.$
>
>
> Since $n_S \approx n \cdot P(S)$, the variance of the estimator decreases at rate $1/n$, even for rare groups. Both asymptotically and for small samples, small subgroups do **not** lead to inflated variance.
>
> That said, the more subtle issue is that in finite samples, the plug‑in variance $\hat{\sigma}^2$ may actually **underestimate** uncertainty when $n_S$ is small. In particular, when the estimated subgroup rate $\hat{p}_S$ is very close to 0 or 1, the term $\hat{p}_S (1 - \hat{p}_S)$ in the variance formula becomes near zero. This drives the Wald variance estimate to be artificially small, even though the underlying sampling uncertainty is high for such extreme proportions with small $n_S$.
>
> This can make the Wald test *anti‑conservative*, falsely rejecting the null due to high variance in small groups not captured by the asymptotic approximation.
>
> This is precisely why we introduce the second test: a Bayesian Dirichlet–multinomial method that provides calibrated inference in the small-sample regime. We will clarify this transition in the paper: the problem is not variance inflation, but **undercoverage** due to inappropriate reliance on asymptotics when subgroup counts are small.
>
> ---
>
> **W3:** We have explored the impact of the prior choice and the incorporation of prior information (derived from the training set) and the revision will include a full analysis contrasting:
>
> * **Uniform (flat) Dirichlet priors**, which impose no substantive prior information;
> * **Empirical‑Bayes priors** derived from training‑set frequencies.  For each subgroup we set
>
> $$
> \boldsymbol\alpha
> =\frac{\lambda}{\max \bigl(1,\min(c_{0,s},c_{0,\bar s},c_{1,s},c_{1,\bar s})\bigr)}
> \bigl[c_{0,s}+1,c_{0,\bar s}+1,c_{1,s}+1,c_{1,\bar s}+1\bigr],
> \qquad \lambda\in\{0.5,1,2,5\},
> $$
>
> where $c_{i,j}$ is the number of training samples with label $i\in\{0,1\}$ in group $j\in\{s,\bar s\}$.
>
> This construction scales the prior strength by $\lambda$ while preserving the empirical class proportions, and caps the influence when any cell count is extremely small.
>
> The following table shows the preliminary results with the confidence intervals (CI) for some selected subgroups from the Adult dataset:
>
> |  | (Other, Under 18, Female) | (A.P.I., Over 65, Male) | (White, Over 65, Female)| (Black) |
> |:---:|---|---|---|---|
> |\$n_S \$| 2 | 12| 210 | 1522 |
> | asymptotic normal CI |[-0.204, -0.192] |[-0.271, 0.041] |[-0.182, -0.123] |[-0.130, -0.097] |
> | Bayesian CI (uniform prior) |[-0.190,  0.488] |[-0.176,  0.176] |[-0.174, -0.115] |[-0.127,  -0.097] |
> | Bayesian CI (empirical-Bayes prior, $\lambda=0.5$) |[-0.240,   0.014] |[-0.196,  0.048] |[-0.179, -0.121] |[-0.138, -0.114] |
> | Bayesian CI (empirical-Bayes prior, $\lambda=1$) |[-0.236,  0.037] |[-0.196,  0.052] |[-0.179, -0.123] |[-0.138, -0.113] |
> | Bayesian CI (empirical-Bayes prior, $\lambda=2$) |[-0.228,   0.035] |[-0.177,  0.037] |[-0.177, -0.125] |[-0.138, -0.114 ] |
> | Bayesian CI (empirical-Bayes prior, $\lambda=5$) |[-0.195, 0.000] |[-0.156,  0.011] |[-0.176, -0.128] |[-0.139, -0.113] |
> | Bootstrapping intervals |[-0.204, -0.192] |[-0.203,  0.073] |[-0.178,  -0.118] |[-0.130, -0.098] |
>
>
> This preliminary sensitivity analysis confirms prior influence for extremely small subgroups, and that all methods rapidly converge to the asymptotic regime as $n_S$ grows. For the smaller $n_S$ case, the Wald intervals are spuriously narrow, clearly underrepresenting the true sampling uncertainty. In contrast, the uniform-prior Bayesian credible intervals are much larger, appropriately reflecting the scarcity of data. As we introduce empirical-Bayes priors, these intervals shrink toward the Wald width but remain more conservative than asymptotic bounds, demonstrating that modest data-informed priors can stabilize inference without overconfidence. With hundreds or thousands of samples, all methods produce nearly identical intervals, illustrating that prior effects vanish once more data is available.
>
> ---
>
> **W4:** Thank you for raising this important point. We experimented extensively with non-parametric bootstrap intervals prior to the submission, and decided not to include them because, in our settings, bootstrap often fails to offer any advantages over either the frequentist or the Bayesian approaches. However, we agree that it would be better to include a discussion about this point in the paper anyway.
> We highlight now some characteristics of the bootstrap approach. First, in moderate to large subgroups, bootstrap intervals converge both theoretically and empirically to the same Wald limits delivered by our analytic CLT. At that scale, bootstrap therefore adds only computational overhead without any improvement. In the small $n_s$ intersectional slices where analytic Wald intervals break down, bootstrap inherits the same weaknesses. Resampling with replacement in class unbalanced scenarios frequently generates samples in which one of the four contingency cells is zero (e.g., no predicted positives for group $S$), producing degenerate or undefined estimates of $SP$. Moreover, the resulting empirical distribution is highly discrete and strongly skewed, which yields uncentered confidence intervals.
>
> In the last row of the table reported in W3, we display the Bootstrapping intervals obtained. One can observe how similar they are to the asymptotic confidence interval, even in the small sample regime, highlighting their inadequate estimation (uncentered with small variance) in such cases.
>  For these reasons, we opted for the analytic CLT, whose validity can be proved, and for the Dirichlet–multinomial credible interval, which remains calibrated even when cell counts are in the single digits. Nonetheless, the camera‑ready version will include a dedicated appendix comparing our method with the bootstrap approach, documenting the pathologies above, and clarifying the regimes in which bootstrap is a viable alternative.
>
> Concerning the lack of experiments with additional metrics,  we have already highlighted it as a limitation. We explained how the framework can be used to compute confidence intervals for some further metrics e.g.  Equal Opportunity, so no extra theory is required; the work is purely computational. Running a full set of new experiments for additional metrics (with several datasets and subgroup configurations) is not realistic within the short rebuttal window, but can be easily done and we plan to do it for a possible camera‑ready version, placing the resulting figures and tables in the appendix. That addition will demonstrate the size‑adaptive behaviour under an additional metric while keeping the main claims unchanged.
>
> ---
>
> We hope these clarifications address the Reviewer’s concerns. Our revisions will add the requested variance discussion, prior-sensitivity experiments, bootstrap comparison, and example extensions to other metrics. We believe these additions highlight the paper’s practical value and methodological novelty. We welcome any follow-up questions and would be happy to provide further details or clarifications.

---

> > ### Comment · Reviewer_L4Jh · 2025-08-03
> > **Response to Rebuttal**
> >
> > Thanks for the detailed rebuttal.
> >
> > I think that the added prior sensitivity analysis and bootstrapping results strengthen the paper. The explanation for how the small subgroups affect the variance estimation is clear to me. One small question regarding the notation for the variance explanation, I think that the equation above holds if $p_S$ denotes $P(f(x)=1|x\in S(\mathcal{X}))$. However, in the paper $p_S$ denotes $P(x\in S(\mathcal{X}))$, a fix there would be appreciated.
> >
> > Considering the prior sensitivity analysis, the comparison with the existing methods such as bootstrapping, I will increase my rating.

---

> ### Author Response · Authors · 2025-08-04
>
> Dear Reviewer,
>
> We thank you again for your questions, which will allow us to improve the paper.
> With respect to the variance expression, the Reviewer is correct, $p̂_S$ indicates $P(f(x)=1|x\in S(\mathcal{X}))$, to be consistent with the notation of the paper,
>
> $p̂_S$
>
> and $p̂_{\bar S}$
>
> should be replaced respectively with
>
> $p̂_{1, S}/(p̂_{1, S}+p̂_{0, S})$
>
> and $p̂_{1, \bar S}/(p̂_{1, \bar S}+p̂_{0, \bar S})$.
>
> We appreciate you catching this inconsistency and are glad our previous response addressed your concerns.

---

> > ### Comment · Reviewer_L4Jh · 2025-08-05
> >
> > Thanks for the clarification regarding the notation, and I agree with the authors on the new notation.

---

### Official Review · Reviewer_W2en · 2025-07-03

**Clarity:** 3
**Significance:** 3
**Originality:** 3
**Rating:** 4
**Confidence:** 4

**Summary:**

This paper proposes a framework for group-size adaptive hypothesis test (SAFT) for auditing a fair classifier. When sufficient samples are available, they propose using a Wald test to get construct a statistically consistent test, and when there is a possibility of very few samples from certain groups, they propose using a Bayesian test with a Dirichlet prior. The advantage over other works lies in the fact that, depending on the subgroup sizes, the proposed testing framework can dynamically adapt.

**Questions:**

My questions are around the three limitations posed previously.
1. Empirical results for other fairness metrics: This can help nail down the effectiveness and "size-adaptiveness".
2. Dependence on minimum support $\tilde{n}$
3. Some justification for how accurate the Bayesian testing framework is.

**Ethical Concerns:**

["NO or VERY MINOR ethics concerns only"]

**Final Justification:**

I am personally not confident with the statistical tools used in this paper, but pretty confident about the problem statement being tackled. After reading other reviews, and due to the fact that conditional parity metrics were not considered (which are very important in fairness literature), but the framework shows promise for working with conditional parity metrics, I will maintain my current rating and will lean towards an Accept.

**Limitations:**

yes

**Paper Formatting Concerns:**

No formatting issues.

**Quality:**

3

**Strengths And Weaknesses:**

Strengths:

1. Solid theoretical results that tackle the said problem very well. The empirical results work very well for SP at least.
2. The Bayesian framework to tackle the small sample regime is a good idea.

Weaknesses:

1. Empirical results for other fairness metrics: The authors admitted in their limitations that this is a pending line of work. Sparse subgroups will have a larger effect on conditional parity metrics like Equal opportunity and Equalized odds, and hence would have been the perfect demonstration of the proposed size-adaptive framework.
2. Dependence on minimum support $\tilde{n}$: The authors fix $\tilde{n}=30$ for switching between the Bayesian inference test and the Wald test. Perhaps an ablation study on the sensitivity of this minimum support parameter can help the reader.
3. While I understand that it is hard for the authors to theoretically validate the effectiveness of the Bayesian inference setup, it will be interesting to check how close it gets to the true hypothesis, perhaps with a simpler data distribution (eg. Gaussian).

---

> ### Author Rebuttal · Authors · 2025-07-30
>
> **W1/Q1**
> We thank the Reviewer for the feedback. We already identify the lack of conditional‑parity experiments as a limitation. We already explained how the  framework can be used to compute confidence intervals for Equal Opportunity and the extention to Equalized Odds is straightforward, so no extra theory is required; the work is purely computational. Running a full set of new experiments for Equal Opportunity and/or Equalized Odds (with several datasets and subgroup configurations) is not realistic within the short rebuttal window, but can be easily done in time for the camera‑ready version, so that the resulting figures and tables can be placed in an appendix. That addition will demonstrate the size‑adaptive behaviour under a conditional metric while keeping the main claims unchanged.
>
> **W2/Q2**
>  The specific cut‑off is a pragmatic choice. Thirty observations is the rule‑of‑thumb widely used to trust CLT‑based Wald tests because coverage and power stabilise around that cell size in typical multinomial settings. Users may raise or lower $\tilde n$  if their data warrant it; the method itself remains unchanged. While an ablation study on the sensitivity of this minimum support parameter is not doable in this short timeframe, we can absolutely launch it and collect/report the results in time for the camera ready version of the paper.
>
> **W3/Q3** To address the reviewer’s request, we performed a simulation study using Gaussian distributions.
> The true Statistical Parity (SP) value (computed from the CDF of the normal) is covered in about **95 %** of cases by the Bayesian posterior credible interval. Coverage is insensitive to sample size, but the interval narrows as more data are used.
>
> We next describe the setup in detail. We consider two groups, $S$ and $S'$, drawn from different normals:
>
> $$
> X_{S} \sim \mathrm{N}(0,1), \qquad
> X_{S'} \sim \mathrm{N}(1,1).
> $$
>
> We define a simple classifier:
>
> $f(x)= 1$  if  $x > 0.5;$
> $f(x)= 0$   otherwise.
>
> For each repetition we sample $n$ points for $S$ and $m$ for $S'$, compute the credible interval for SP, and check if it contains the true value. We repeat this **1000** times. The true statistical parity value can be computed analytically from the cumulative density function of the Gaussian distributions and it is **SP(S)=0.383**.
>
> The credible interval hits its nominal **95\%** coverage for any $n,m$. As $n,m$ grow, the interval tightens around the true SP. With $n=m\in\{5,10,30,100,1000\}$, the last‑run intervals are:
>
> | *n = m* | *Credible Interval* | *Pr*(0 ∈ CI) |
> | ---: | ---: | ---:|
> | 5       | (−0.020, 0.797)     | 0.85         |
> | 10      | (0.028, 0.635)      | 0.61         |
> | 30      | (0.175, 0.616)      | 0.12         |
> | 100     | (0.285, 0.533)      | 0.00         |
> | 1000    | (0.352, 0.433)      | 0.00         |
>
> *Bayesian credible intervals for different sample sizes; the last column is the empirical probability (over 1000 runs) that the interval contains 0.*
>
> Larger samples quickly force the interval away from 0, enabling confident detection of fairness violations.
>
> We would be grateful to engage the Reviewer in a fruitful discussion and to provide further details if needed.

---

> > ### Comment · Reviewer_W2en · 2025-08-03
> > **Response to Rebuttal**
> >
> > I thank the authors for providing a detailed rebuttal.
> >
> > Q1/W1: Forgive me if I did not catch this from the main paper, but why would your results still be effective for conditional error rate metrics? In my understanding (from standard fairness auditing papers), the number of samples required would increase dramatically. Also, I understand that doing the experiments is unrealistic during this short window.
> >
> > Q3/W3: Thank you for the simulation experiments.
> >
> > Currently, I am satisfied with some of the answers provided. I will maintain my current rating for now and wait for more discussion and the end of the discussion period.

---

> > > ### Author Response · Authors · 2025-08-03
> > > **Response to the follow-up on conditional-parity metrics**
> > >
> > > Thank you for giving us the opportunity to clarify.
> > >
> > > Our statement that "no extra theory is required" refers to the **derivation** of a valid test statistic and its variance, not to the power that the resulting test will enjoy in practice.
> > >
> > > For conditional-error–rate metrics such as Equal Opportunity (EO) we restrict attention to the subsample with $Y=1$ and compare $P\bigl(f(X)=1\mid Y=1,S=s\bigr)$ versus $P\bigl(f(X)=1\mid Y=1,S=\bar s\bigr)$.  Technically, this is the same four-cell framework we use for Statistical Parity, so the CLT + Delta and the Bayesian Dirichlet–multinomial methods don't change. Hence the **validity** of our intervals and tests is preserved.
> > >
> > > What does change, exactly as the reviewer notes, is the **effective sample size**: we now operate on the (often much smaller) set of positives.  Our size-adaptive procedure makes this issue explicit rather than hiding it.  If the EO counts for subgroup $s$ contains only a small number of positive examples, the Bayesian interval will widen and the frequentist Wald test will give place to the Bayesian mode, signalling that *no statistically defensible fairness claim can be made*.  Conversely, if enough positives are present, the intervals narrow and the test regains power.  In this sense the framework remains "effective": it prevents overconfident conclusions when data are insufficient and exploits asymptotics when they are.
> > >
> > > This is an interesting discussion  and we agree that this behaviour is itself informative and worth discussing in the paper, since it would have been a good demonstration of the proposed size-adaptive framework.
> > > We hope this clarifies the conceptual point: the methodology generalises seamlessly, but the sample-size requirements inherent to conditional metrics are made visible, rather than obscured, by our approach.

---

> > > > ### Comment · Reviewer_W2en · 2025-08-06
> > > >
> > > > I appreciate the authors being upfront about the limitations of their work. I am satisfied with the responses for now and will maintain my current rating.

---

### Official Review · Reviewer_sptv · 2025-07-03

**Clarity:** 2
**Significance:** 3
**Originality:** 3
**Rating:** 5
**Confidence:** 4

**Summary:**

When we look at intersections of protected classes (i.e. using an intersectional lens) then the size of the intersectional group is smaller and sometimes too small to allow for reliable testing of fairness measures such as disparate impact. The authors wish to find the granularity/resolution at which a given data set can be assessed for fairness/unfairness.

As I understand it, the goal is to provide different levels of "fairness" to intersectional groups based on their size. This can allow more fine-grained control of type I/type II error when testing/auditing for unfairness.

**Questions:**

* One challenge in fairness research is that there are so many notions of fairness, each with their embedded assumptions: a nice exposition of this was given a while ago by Arvind Narayanan in a tutorial [21 fairness definitions and their politics](https://www.youtube.com/watch?v=jIXIuYdnyyk). How many of these (beyond the 3 mentioned in the paper) fall under Theorem 2?

* One of the stated contributions is that many fairness measures have "pitfalls" with respect to intersectionality. However, I think many of the referenced papers also point out such issues with prior fairness measures. The fact that any threshold test will have false positives and false negatives is pretty obvious, so what do the authors mean by this contribution?

* (line 47) Is the right question whether we *should* rigorously guarantee fairness or whether we *can* rigorously guarantee fairness?

* I found the discussion and framing of this a bit confrontational with respect to prior work. Of course, checking for unfairness is a kind of (hypothesis) test: one computes a test statistic and compares it to a threshold. They claim prior measures *require* a practitioner to set a global threshold without looking at the particulars of the data set. From Figure 4 in the submitted manuscript, it seems that a linear correction for group size on $\gamma$-SP would allow for a fixed threshold.

* My understanding is that all of the distributions under consideration are PMFs or conditional PMFs. The estimators are plug-in estimates based on the empirical joint PMF and (as shown in the appendix). Obviously, you have to compute the covariances etc. to get a full characterization of the asymptotics, but from my reading Theorem 2 showing asymptotic normality is basically a kind of textbook application of the delta method: am I missing something here?

* Is it really "rigorous" to set thresholds based on the thresholds given by the asymptotics? This is the large sample test and is the general recipe used in statistical practice but the type I and type II error guarantees are then only approximate, right?

* For the Bayesian setting, I do like the idea of incorporating prior information into the prior. Is it possible to do an experiment exploring the impact of prior information/bias?

* I feel like the word "rigorous" in the paper does a lot of heavy lifting. In Algorithm 1, what is the "rigorous" justification for $\tilde{n} = 30$? Similarly, what $K$ was used for the Bayesian method?

* Apart from conforming to arbitrary norms for error tradeoffs, why select $\alpha = 0.05$? Shouldn't this problem be looked at in more general Neyman-Pearson tradeoff sense?


**Smaller comments:**

There are some other works on addressing intersectionality in ML + fairness: the authors may be interested in seeing how those fit in within the landscape of the literature review:

* Ghosh et al, [Characterizing Intersectional Group Fairness with Worst-Case Comparisons](https://proceedings.mlr.press/v142/ghosh21a.html), 2021.
* Foulds et al, [Differential Fairness: An Intersectional Framework for Fair AI](https://www.mdpi.com/1099-4300/25/4/660), 2023. (Maybe subsumes reference [12]?)
* Himmelreich et al. [The Intersectionality Problem for Algorithmic Fairness](https://arxiv.org/pdf/2411.02569), 2024. (This was posted close to the submission deadline, however).

**Ethical Concerns:**

["NO or VERY MINOR ethics concerns only"]

**Limitations:**

Yes.

**Paper Formatting Concerns:**

N/A.

**Quality:**

3

**Strengths And Weaknesses:**

**Strengths:**

* Proposes a framework for taking into account sample size when testing/auditing for fairness in small subgroups.
* Extensive experiments help illustrate the results.

**Weaknesses:**

* The presentation of the results obscures the main technical tool, which is a straightforward result about the asymptotic normality of statistics calculated from plug-in estimates of discrete random variables.
* The promised "principled methodology" also contains some arbitrary parameter choices.

---

> ### Author Rebuttal · Authors · 2025-07-30
>
> We thank the Reviewer for their positive scores, their in-depth understanding of our work, and their appropriate comments, which will be useful to improve the presentation of our work. We next answer the reviewer's questions.
>
> ---
>
> ## Questions:
>
> > **Q1:** *[...] many notions of fairness [...]*
>
> Unfortunately, the cited tutorial lists definitions only in prose and offers no mathematical formulation (see slide 2 "Limitations of this tutorial"), and we couldn't find supporting documents for the tutorial. To answer the Reviewer's question about which definitions can be modeled by our famework, we refer to the paper *'Fairness definition explained'* [Verma and Rubin, 2018]. Our framework can model the first 13 definitions of table 1, while the last 7 are excluded. Essentially, our framework can model all those based on the confusion matrix or related conditional probabilities.
>
> ---
>
> > **Q2:** *[...] many fairness measures have "pitfalls" with respect to intersectionality [...]*
>
> Precisely. Because most metrics were not designed for sparse, intersectional subgroups, fixed‑threshold tests often suffer high false‑positive and false‑negative rates. Our contribution is to quantify when and how severe these failures become, and to provide size‑adaptive tests that remain reliable in such settings. Furthermore, prior fairness metrics based on thresholds do not have Type-I/II error guarantees.
>
> ---
> > **Q3:** *[...] or whether we can rigorously guarantee fairness?*
>
> Thank you for the clarification. We agree that “can” is more appropriate, given that our paper focuses on testing for fairness. We will adopt this wording in the revision.
>
> ---
> > **Q4:** *[...] would allow for a fixed threshold.*
>
> A linear adjustment to γ‑SP has several limitations. There is no principled way to choose its slope or intercept, it comes with no statistical guarantees, and, as Figure 4 shows, even the best‑tuned linear correction would still misclassify points that our method separates correctly.
>
> ---
> > **Q5:** *[...] Theorem 2 showing asymptotic normality [...]*
>
> The reviewer is correct: Theorem 2 is obtained by applying the multivariate CLT, the delta method and Slutsky’s theorem. One interesting technical choice was to express the fairness metrics as functions of probabilities of disjoint events. This keeps the covariance matrix analytic and easily computable; without that disjointness the algebra would be considerably heavier. The novelty is therefore not the asymptotic argument itself but (i) showing that the resulting Wald test slots seamlessly into our size‑adaptive framework and (ii) pairing it with a calibrated Bayesian test that covers the small‑sample regime where the asymptotics are unreliable. We will add a short remark to the camera ready version of the manuscript to make this clearer.
>
> ---
> > **Q6:** *Is it really "rigorous" to set thresholds [...]*
>
>
> In finite-sample inference, all frequentist tests based on sampling distributions are approximate unless the distribution is exactly known. For large groups, our Wald test is derived from a formal CLT with a plug‑in variance estimator. This is not an ad-hoc correction: under standard regularity conditions, the rejection region is calibrated to control type I error at level α as n→∞, which is the definition of an asymptotically valid test. By the multivariate CLT and delta method, we obtain a limiting normal distribution with an explicit variance estimator. This ensures that, as n grows, the rejection region converges to one with exact type I error α. In other words, the α‑level control is not heuristic but a formal consequence of the CLT under standard regularity conditions. Moreover, this is the same notion of “rigor” used in essentially all classical hypothesis tests (e.g., t‑tests, χ² tests, Wald tests in GLMs). Calling it approximate is correct in a finite-sample sense, but asymptotic tests are considered rigorous because their error properties are analytically characterized and converge to the nominal level. In contrast, fixed-threshold fairness audits offer no such convergence guarantee at all - their type I/type II error is uncontrolled even asymptotically.
>
> ---
> > **Q7:** *[...] the impact of prior information/bias?*
>
>
> We have explored the impact of the prior choice and the incorporation of prior information (derived from the training set) and the revision will include a full analysis contrasting:
>
> * **Uniform (flat) Dirichlet priors**, which impose no substantive prior information;
> * **Empirical‑Bayes priors** derived from training‑set frequencies.  For each subgroup we set
>
> $$
> \boldsymbol\alpha
> =\frac{\lambda}{\max \bigl(1,\min(c_{0,s},c_{0,\bar s},c_{1,s},c_{1,\bar s})\bigr)}
> \bigl[c_{0,s}+1,c_{0,\bar s}+1,c_{1,s}+1,c_{1,\bar s}+1\bigr],
> \qquad \lambda\in\{0.5,1,2,5\},
> $$
>
> where $c_{i,j}$ is the number of training samples with label $i\in\{0,1\}$ in group $j\in\{s,\bar s\}$.
>
> This construction scales the prior strength by $\lambda$ while preserving the empirical class proportions, and caps the influence when any cell count is extremely small.
>
> The following table shows the preliminary results with the confidence intervals (CI) for some selected subgroups from the Adult dataset:
>
> |  | (Other, Under 18, Female) | (A.P.I., Over 65, Male) | (White, Over 65, Female)| (Black) |
> |:---:|---|---|---|---|
> |\$n_S \$| 2 | 12| 210 | 1522 |
> | asymptotic normal CI |[-0.204, -0.192] |[-0.271, 0.041] |[-0.182, -0.123] |[-0.130, -0.097] |
> | Bayesian CI (uniform prior) |[-0.190,  0.488] |[-0.176,  0.176] |[-0.174, -0.115] |[-0.127,  -0.097] |
> | Bayesian CI (empirical-Bayes prior, $\lambda=0.5$) |[-0.240,   0.014] |[-0.196,  0.048] |[-0.179, -0.121] |[-0.138, -0.114] |
> | Bayesian CI (empirical-Bayes prior, $\lambda=1$) |[-0.236,  0.037] |[-0.196,  0.052] |[-0.179, -0.123] |[-0.138, -0.113] |
> | Bayesian CI (empirical-Bayes prior, $\lambda=2$) |[-0.228,   0.035] |[-0.177,  0.037] |[-0.177, -0.125] |[-0.138, -0.114 ] |
> | Bayesian CI (empirical-Bayes prior, $\lambda=5$) |[-0.195, 0.000] |[-0.156,  0.011] |[-0.176, -0.128] |[-0.139, -0.113] |
>
> This preliminary sensitivity analysis confirms prior influence for extremely small subgroups, and that all methods rapidly converge to the asymptotic regime as $n_S$ grows. For the smaller $n_S$ case, the Wald intervals are spuriously narrow, clearly underrepresenting the true sampling uncertainty. In contrast, the uniform-prior Bayesian credible intervals are much larger, appropriately reflecting the scarcity of data. As we introduce empirical-Bayes priors, these intervals shrink toward the Wald width but remain more conservative than asymptotic bounds, demonstrating that modest data-informed priors can stabilize inference without overconfidence. With hundreds or thousands of samples, all methods produce nearly identical intervals, illustrating that prior effects vanish once more data is available.
>
> ---
> > **Q8:** *I feel like the word "rigorous" [...]*
>
> The testing framework is rigorous; the specific cut‑off is a pragmatic choice. Thirty observations is the rule‑of‑thumb widely used to trust CLT‑based Wald tests because coverage and power stabilise around that cell size in typical multinomial settings. Users may raise or lower \tilde n if their data warrant it; the method itself remains unchanged.
> For the Bayesian method, we set K = 10.000. Ten‑thousand Dirichlet-multinomial samples keep Monte‑Carlo noise below 0.2 pp at the 5 % level while remaining computationally trivial. Any larger K only reduces that already‑negligible error; smaller K risks noticeable numerical variance. Also in this case, practitioners can calibrate K to their precision/speed trade‑off.
>
> ---
> > **Q9:** *[...] why select alpha = 0.05? [...]*
>
> Our test is built exactly in the Neyman–Pearson framework: for every subgroup $S$, we define $H0:SP(S)=0$ and a rejection region calibrated to control type I error at level $\alpha$. The choice of $\alpha=0.05$ in the experiments is not a claim of optimality; it is a standard convention to pick one operating point on the type I/type II error curve for comparison with prior fairness audits.
> More importantly, the method is agnostic to the specific $\alpha$. For any $\alpha \in (0,1)$, the procedure yields a valid $α$‑level test (asymptotically for the Wald version, exactly for the Bayesian small-sample version). By sweeping α, one can trace out the full ROC‑like tradeoff between false fairness violations (type I) and missed unfairness (type II). This is the general Neyman–Pearson sense: we provide a test statistic with a calibrated null distribution, and practitioners or regulators can set α based on the cost of false positives vs. false negatives.
>
> In short, $\alpha=0.05$ is a reporting choice, not a methodological constraint; the framework itself already encodes the Neyman–Pearson tradeoff and supports any α the user deems appropriate. We can make this explicit in the main text.
>
> Furthermore, to strengthen the connection with the Neyman–Pearson lemma it should be possible to define a likelihood ratio test, which is the uniformly most powerful test in the set of level alpha tests. We will consider this extension in a possible camera ready version.
>
> ---
> > **Q10:** *There are some other works on addressing intersectionality in ML + fairness [...]*
>
> We are grateful to the Reviewer for pointing out these related papers: we will add them to our literature review.
>
> ---
> ---
> **We welcome any follow‑up questions and would be happy to provide additional details or clarifications.**

---

> > ### Comment · Reviewer_sptv · 2025-08-03
> > **Thank you for your response**
> >
> > Thanks for the detailed responses. I think we differ in the notion of what is reasonable practice. Just because a lot of statistical practice substitutes asymptotics to set thresholds just means the thresholds are "good enough for practice" but in fact do not conform to finite sample guarantees. But whatever, we could spend an eternity running back and forth on this: maybe it's best for me to go back and read Breiman's two cultures paper and the like.
> >
> > I think emphasizing in the revision that this works for anything using the confusion matrix would be particularly helpful.

---

### Note · Authors · 2025-08-15

We are pleased with the clear consensus among reviewers on the paper's clarity, technical soundness, and practical significance. Reviewers acknowledged that unifying a CLT-based Wald test for large subgroups with a calibrated Bayesian test for sparse ones provides a novel and principled framework for size-adaptive fairness auditing - an approach absent from current literature. During the discussion, the reviewers requested a deeper analysis of key methodological choices; we have addressed these points with new experiments and clarifications.

We thank all reviewers for their constructive feedback, which will help us further strengthen the camera-ready version.

As noted in the discussion, the remaining disagreement with Reviewer ujf4 is not about the correctness of our work, but rather a good-faith difference in perspective on what constitutes a novel contribution for NeurIPS. In this regard, we refer to the NeurIPS 2025 Reviewer Guidelines. Given the overall feedback from the reviewers, we hope the AC will weigh the significance of this practical and methodological contribution against this single reviewer's differing perspective on theoretical novelty.

---

### Decision · Program_Chairs · 2025-09-17

**Decision:**

Accept (poster)

**Comment:**

The paper tackles the important problem of fairness auditing in machine learning using a size-adaptive hypothesis testing framework that addresses the statistical challenges of evaluating fairness metrics across demographic subgroups of varying sizes. The authors propose a unified approach that switches between a Central Limit Theorem-based Wald test for large subgroups and a Bayesian Dirichlet-multinomial method for small intersectional groups, aiming to replace ad-hoc threshold-based fairness assessments with statistically rigorous hypothesis tests that provide type-I error guarantees.

The reviewers and authors engaged in substantive discussions across multiple dimensions. Reviewer sptv gave an accept rating (5) but raised technical questions about the scope of fairness metrics covered by the framework and challenged the characterization of statistical "rigor," noting that asymptotic tests provide approximate rather than exact guarantees. Reviewer W2en gave a borderline accept rating (4) and requested empirical validation for conditional fairness metrics and sensitivity analysis for the minimum support parameter, leading to productive exchanges about the framework's generalizability. Reviewer L4Jh gave a borderline accept rating (4) and questioned the technical novelty and requested comparisons with bootstrap methods, prompting the authors to provide detailed analysis showing bootstrap's limitations in sparse data regimes. Most critically, Reviewer ujF4 gave a borderline reject rating (3) and argued that the theoretical contributions were straightforward applications of standard statistical methods, representing more of an educational note than a novel research contribution, though they acknowledged the paper's clarity and technical soundness.

Based on the review discussions, I recommend acceptance because the work addresses a genuine methodological gap in fairness auditing practice. While Reviewer ujF4 correctly notes that the individual statistical techniques are well-established, the unified framework that adapts between asymptotic and Bayesian approaches based on sample size represents a meaningful practical contribution to the fairness-in-ML community. The authors gave detailed rebuttals that existing approaches like bootstrap methods fail in precisely the sparse intersectional settings where their method provides value. The clear consensus among three of four reviewers on the paper's technical soundness, practical utility, and addressing of an important problem outweighs concerns about theoretical novelty, particularly given NeurIPS's stated value for work that bridges theory and practice in societally critical applications.